# CS-pFedTM: Communication-Efficient and Similarity-based Personalized Federated Learning with Tsetlin Machine

## Abstract

Federated Learning has become a promising framework for preserving data privacy in collaborative training across decentralized data sources. However, the presence of data heterogeneity remains a significant challenge, impacting both the performance and efficiency of FL systems. To address this, we introduce CS-pFedTM (Communication-Efficient and Similarity-based Personalized Federated Learning with Tsetlin Machine), a method that addresses this challenge by jointly enforcing communication-aware resource allocation and heterogeneity-driven personalization. CS-pFedTM enforces communication budget feasibility through clause allocation and tailor personalization using clients' parameters similarity as a proxy for data heterogeneity. Experiments across multiple datasets show that CS-pFedTM consistently outperforms state-of-the-art personalized FL approaches, achieving at least $3.6\times$ lower upload cost, $5.58\times$ lower download cost, and $1.17\times$ higher runtime efficiency, while maintaining superior accuracy.

## 1 Introduction

Federated Learning (FL) enables clients to train models locally while only sharing parameters, preserving privacy as sensitive data remain on individual devices (McMahan et al., 2016). Despite its promise, FL still faces two major challenges: data heterogeneity across clients and communication constraints, which bottleneck scalability in real-world systems (Khan et al., 2021).

Personalized FL addresses data heterogeneity by combining locally adapted models with shared global knowledge. The central challenge in this lies in balancing effective personalization with communication efficiency. Existing methods partially tackle this trade-off but often lack the ability to provide adaptable, fine-grained personalization and flexible control over communication costs (Shamsian et al., 2021; Gohari et al., 2024). Furthermore, most approaches rely on deep neural networks (DNNs) (Asad et al., 2023; Lei et al., 2020), which incur high computational and memory costs, limiting their practicality for resource-constrained edge devices (Almanifi et al., 2023; Khan et al., 2021).

To overcome these limitations, we leverage the low-complexity Tsetlin Machine (TM), a rule-based model based on finite-state automata and game theory, as an efficient alternative to DNNs (Lei et al., 2020; 2021). We propose CS-pFedTM (Communication-Efficient, Similarity-based Personalized FL with TM), which simultaneously addresses data heterogeneity and communication efficiency. Our analysis reveals a strong correlation between TM clause parameters and the underlying FL data distribution, motivating personalization based on data heterogeneity. Our method also accounts for communication budgets when allocating clause contributions, and incorporates weight masking to handle locally absent classes to optimize performance and efficiency. Our approach improves runtime efficiency by at least $1.17\times$, respectively, while reducing upload communication by $3.6\text{--}886\times$ and download communication by $5.58-107\times$ compared to state-of-the-art (SOTA) communication-efficient personalized FL baselines.

In summary, our contributions are as follows:

- We introduce a novel TM-based personalization scheme in which each client trains both a local and a global model, while communicating only the global model. To improve

flexibility and efficiency, we incorporate class-specific weight masking and performance-based client selection, all without requiring clients to share metadata.

- We show that the similarity between clients' TM parameters reflects overall data heterogeneity, which we exploit to adaptively allocate local and global clauses. Higher heterogeneity leads to more local clauses to strengthen personalization, while lower heterogeneity shifts the balance toward global clauses to reinforce shared knowledge.

- We proposed a a budget-constrained allocation mechanism that adjusts this allocation according to communication limits, supporting efficient and adaptive personalization.

- Extensive experiments show that CS-pFedTM outperforms SOTA communication-efficient personalized FL baselines while significantly reducing communication, storage, runtime, and training latency.

## 2 RELATED WORK

In FL, data heterogeneity and communication efficiency are major challenges (Tan et al., 2023; Asad et al., 2023). Strategies such as quantization (Mao et al., 2022; Reisizadeh et al., 2019; Hönig et al., 2022), sparsification (Qiu et al., 2022; Rothchild et al., 2020), and network pruning (Jiang et al., 2022; Li et al., 2021) reduce communication and computation. Alternative architectures such as Binary Neural Networks (BNN) (Yang et al., 2021) and Tsetlin Machines (TM) (How et al., 2023) further reduce the size and memory of the model, improving efficiency.

Beyond efficiency, substantial progress has been made in addressing data heterogeneity in FL (Imteaj et al., 2022; Tan et al., 2023; Fallah et al., 2020). Multi-task learning (T. Dinh et al., 2020; Smith et al., 2017) couples client-specific models with a global representation, meta-learning (Fallah et al., 2020; Jiang et al., 2023) enables rapid local adaptation, clustering (Sattler et al., 2021) groups similar clients, and knowledge distillation (Li & Wang, 2019) transfers knowledge via teacher–student frameworks. Personalization via latent distribution modeling (Marfoq et al., 2022; Mclaughlin & Su, 2024) explicitly captures data variability, balancing local flexibility and global generalization.

A complementary line of work simultaneously tackles personalization and communication efficiency. Parameter decoupling methods such as LG-FedAvg, FedRep, FedBABU, FedPer, and Fed-PAC (Liang et al., 2020; Collins et al., 2023; Oh et al., 2022; Arivazhagan et al., 2019; Xu et al., 2023) separate client-specific and global components but remain coarse-grained and fixed. Fed-Select (Tamirisa et al., 2024), inspired by the Lottery Ticket Hypothesis, discovers fine-grained subnetworks via parameter masks, though fairness concerns arise since non-selected clients do not benefit from aggregation. Similarly, sparsification-based personalization methods such as DisPFL (Dai et al., 2022), a decentralized FL method, prune dynamically to exchange only active weights between clients, and SpaFL (Kim et al., 2024) communicates only trainable thresholds, reducing communication by two orders of magnitude. While effective, these approaches still impose structural constraints and do not adaptively allocate shared versus local parameters based on client heterogeneity.

TM-based FL methods such as FedTM (How et al., 2023) do not address data heterogeneity, while the more recent Tsetlin-Personalized Federated Learning (TPFL) (Gohari et al., 2024) introduces personalization through confidence-based clustering, aggregating clients within clusters that share similar class-wise confidence profiles. Although TPFL incorporates a form of personalization, it does not adaptively adjust the balance between local and global TM components, nor does it consider communication constraints in the personalization process.

## 3 BACKGROUND

### 3.1 TSETLIN MACHINE

TM is a machine learning algorithm that employs propositional logic to capture frequent patterns. It operates using Tsetlin Automata (TA) arranged in teams, building discriminative conjunctive clauses and utilizing a majority voting mechanism for final classification (Granmo, 2021).

### 3.1.1 TSETLIN MACHINE STRUCTURE

The TM structure is based on a two-action TA, building upon reinforcement learning principles.

Consider an input vector of $o$ propositional variables: $\mathbf{x} = \{x_1, \ldots, x_o\} \in \{0,1\}^o$. Along with their negated counterparts, $\{\neg x_1, \ldots, \neg x_o\}$, the variables together form a literal set $L = \{l_1, \ldots, l_{2o}\} = \{x_1, \ldots, x_o, \neg x_1, \ldots, \neg x_o\}$. The TM comprehends the structure of each conjunctive clause ($C_j(\mathbf{x})$), indexed by $j$, by defining its literals through a team of $2o$ TAs. A conjunctive clause is constructed by taking the AND operation of a subset $L_j \subseteq L$:

$$C_j(\mathbf{x}) = \bigwedge_{l_k \in L_j} l_k.$$

With $n$ clauses and $2o$ literals, we have $2o \cdot n$ TAs. Each TA makes decisions on whether to exclude or include the associated literal in the conjunctive clause.

### 3.1.2 TSETLIN MACHINE LEARNING MECHANISM

TM learning begins by converting training data into boolean form, enabling the creation of conjunctive clauses from literals (input variables and their negations). For $n$ clauses, $n/2$ positive clauses identify class $y = 1$, and $n/2$ negative clauses identify class $y = 0$. Training occurs online, processing one example $(\mathbf{x}, y)$ at a time.

Using $(\mathbf{x}, y)$, the TM adjusts its TAs via two feedback types, which decide whether input literals should be included in clauses that vote for a class. Type I Feedback strengthens clauses corresponding to the correct class, increasing the chance of outputting 1, while Type II Feedback suppresses clauses that would cause false positives. Feedback is applied to a random subset of clauses, controlled by hyperparameter $T$, so that the sum $s(\mathbf{x}) = \sum_{j=1}^{n/2} C_j^+(\mathbf{x}) - \sum_{j=n/2+1}^{n} C_j^-(\mathbf{x})$, approach $-T$ for $y = 0$ or $T$ for $y = 1$. The sum is clamped, and feedback probabilities are proportional to the difference between the clamped sum, $c(\mathbf{x}) = \text{clamp}(s(\mathbf{x}), -T, T)$, and the target.

$$p_y(\mathbf{x}) = \begin{cases} \frac{T + c(\mathbf{x})}{2T}, & \text{if } y = 0 \\ \frac{T - c(\mathbf{x})}{2T}, & \text{if } y = 1 \end{cases} \tag{1}$$

The randomized selection of clauses ensures diverse feedback distribution, preventing clustering on specific patterns and fostering recognition across various sub-patterns. In essence, TM's learning mechanism refines clause evaluations over successive training cycles, adapting to specific class objectives and promoting effective pattern recognition.

**Weighted TM:** The introduction of weights entails assigning positive real-valued weights to individual clauses, facilitating a more concise representation of the clause collection. By adjusting these weights, the influence of particular clauses can be altered, contributing to a real-valued overall sum within the TM (Phoulady et al., 2020). The resulting overall sum, denoted as $s(\mathbf{x})$, becomes a real-valued quantity: $s(\mathbf{x}) = \sum_{j=1}^{n/2} w_j^+ C_j^+(\mathbf{x}) - \sum_{j=n/2+1}^{n} w_j^- C_j^-(\mathbf{x})$

**Multi-Class TM:** For classification, the TM applies the unit step function to the sum ($u(s(\mathbf{x}))$). If the signed sum is negative, the TM outputs $y = 0$; otherwise, it outputs $y = 1$. In the multi-class scenario, it adheres to a comparable operational pattern. Each class, denoted as $m = 1, ..., M$, possesses its own TA teams. Suppose the current observation $(\mathbf{x}, y)$ has $y = k$, the TA teams affiliated with class $k$ are trained as $y = 1$. Concurrently, a random class $l \neq k$ is selected and the TA teams associated with class $l$ are then trained as $y = 0$. In this scenario, the threshold function for each output $y$ is modified by utilizing the $\arg\max$ operator to output the class $m$ that corresponds to the largest sum, $s^m(\mathbf{x}) = \sum_{j=1}^{n/2} w_j^{+,m} C_j^{+,m}(\mathbf{x}) - \sum_{j=n/2+1}^{n} w_j^{-,m} C_j^{-,m}(\mathbf{x})$, to determine the final output of the TM:

$$\hat{y} = \underset{m=1...M}{\arg\max} \, s^m(\mathbf{x}), \tag{2}$$

**Convolutional TM (CTM):** Inspired by convolutional structures in DNNs, filters with spatial dimensions $W \times W$ and $Z$ binary layers are utilized. Each image, with dimensions $X \times Y$ and $Z$

binary layers is modeled in TMs using an input vector $\mathbf{x} = \{x_k \mid k \in \{0,1\}^{X \times Y \times Z}\}$. In CTM, clauses function as filters, each composed of $X \times Y \times Z \times 2$ literals (Granmo et al., 2019).

In the CTM, the input vector represents an image patch, and an image contains $B$ patches. There are $B$ literal inputs per clause. Each clause outputs $B$ values per image (one value per patch) instead of a single output for the TM. The output of a positive clause $j$ on patch $b$ is denoted as $c_j^b$. To consolidate multiple outputs $c_j^1, \ldots, c_j^B$ of clause $j$ into a single output $c_j$, a logical OR operation is applied: $c_j = \bigvee_{b=1}^{B} c_j^b$. Training builds upon the learning process of TM, encompassing Type I and Type II feedback. To determine which patch to use during clause updating, the CTM randomly selects a single patch from those contributing to the clause evaluating to 1. The clause is then updated based on this chosen patch.

**TM Composites:** TM Composites, as introduced in Granmo (2023), foster collaboration among multiple independently trained TM models. Instead of utilizing the $\arg\max$ operator as described in Equation 2, to determine the class index $m$ associated with the largest sum, TM composites involve computing the class sums, $s_t^m(\mathbf{x})$, for each TM $t$, where $t \in \{1, 2, ..., r\}$. These class sums are then normalized by dividing by the difference between the maximum and minimum class sums in the input set, $(\alpha_t = \max_m(s_t^m(\mathbf{x})) - \min_m(s_t^m(\mathbf{x})))$.

The final class output is determined by the maximum value of the sum of all $r$ TMs, calculated as:

$$\hat{y} = \arg\max_m \left( \sum_{t=1}^{r} \frac{1}{\alpha_t} s_t^m(\mathbf{x}) \right) \tag{3}$$

## 4 METHODOLOGY

Before presenting the full method, we first introduce our novel personalization scheme in CS-pFedTM, which addresses limitations in TM-based FL approaches in handling data heterogeneity (How et al., 2023). Building on this scheme, CS-pFedTM jointly adapts global and local clause allocations based on client heterogeneity and communication constraints, achieving an optimal balance between personalization and efficiency.

### 4.1 PERSONALIZATION

Our personalization strategy improves the adaptability of the local model to client-specific data while leveraging global knowledge. Each client maintains two independent TMs: a local TM, trained exclusively on its own data to capture client-specific patterns, and a global TM, also trained locally but whose parameters are shared with the server. During each communication round, only the global TM parameters are uploaded to the server; the server aggregates these updates and returns the updated global model to clients.

Clients then combine the outputs of the local and global TMs using Equation 3, integrating local adaptation and shared global knowledge. Furthermore, the class-specific weights of TMs allow for further personalization through weight masking: weights corresponding to classes not observed locally can be set to zero, enabling the model to quickly adapt to unseen classes. This design ensures robust and flexible personalization in FL with heterogeneous data.

### 4.2 PROBLEM FORMULATION

While this personalization framework enables clients to adapt effectively to heterogeneous data, the allocation of clauses between local and global components directly impacts both performance and efficiency. Clients with more heterogeneous data benefit from a larger fraction of local clauses to capture client-specific patterns, whereas clients with less heterogeneous data can rely more on global clauses for shared knowledge Additionally, communication constraints impose upper limits on the amount of information each client can share per round.

The challenge, therefore, is to determine the optimal allocation of local and global clauses that maximizes performance while adhering to defined communication budgets, without requiring clients to share explicit metadata about their data distributions. This motivates CS-pFedTM, our

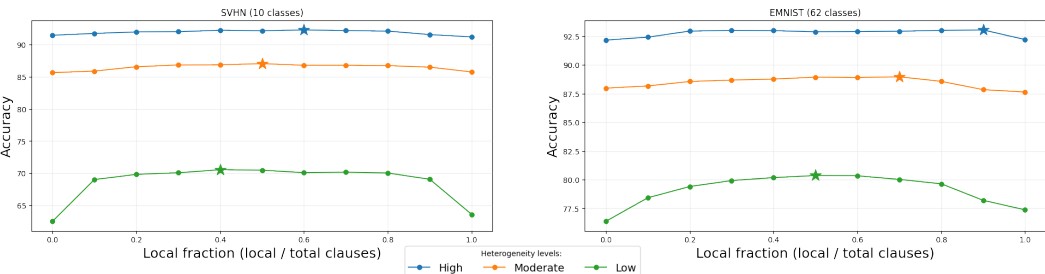

Figure 1: Effect of local clause fraction on performance. Peak performance shifts to higher fractions with increasing heterogeneity and class count

communication-efficient personalization framework, which leverages the similarity of trained TM parameters across clients to guide adaptive clause allocation.

### 4.2.1 EFFECT OF DATA HETEROGENEITY ON PERSONALIZATION

We first study fixed local-global splits to understand performance trends. As shown in Figure 1, performance consistently degrades at both extremes: allocating nearly all clauses locally or globally leads to suboptimal outcomes. Instead, peak performance emerges at intermediate allocations. For highly heterogeneous clients, retaining more local clauses improves personalization, and similarly, datasets with a larger number of classes also require a higher fraction of local clauses to reach peak accuracy. This occurs because higher heterogeneity and increased number of classes enhances the diversity of patterns each client must capture locally, making a larger fraction of local clauses necessary to model client-specific distributions effectively.

This shows that no fixed allocation is optimal across all heterogeneity levels, motivating our adaptive allocation mechanism that dynamically adjusts the local-to-global ratio based on heterogeneity.

### 4.2.2 EXPLORING THE CONNECTION BETWEEN TRAINED PARAMETERS AND DISTRIBUTION DISTANCES

TMs are sensitive to data distributions due to stochastic clause updates and clauses corresponding to underrepresented patterns tend to be reinforced less (Granmo, 2021). As a result, the learned clauses encode the statistical properties of the training data. In FL, this implies that clients with heterogeneous data produces distinct TM parameters, naturally reflecting differences in local distributions.

We show that parameter similarity across clients inversely reflects data heterogeneity: Clients with high data heterogeneity exhibit lower parameter similarity, while less heterogeneous clients yield higher parameter similarity. Let $W(q_A q_B)$ denote the Wasserstein distance between two data distributions, and $\mathcal{J}(S_A, S_B)$ the Jaccard similarity between their trained TM parameters, which quantifies the overlap of active clauses between models trained on the different distributions.

**Corollary 1 (Inverse Relation Between Distribution Divergence and Clause Overlap)** *Let* $q_A$ *and* $q_B$ *be two class distributions and* $S_A$, $S_B$ *be the corresponding trained TM states (sets of clauses). Then:*

$$W(q_A, q_B) \longrightarrow smaller \implies \mathcal{J}(S_A, S_B) \longrightarrow larger,$$

*Thus, lower distributional divergence corresponds to higher parameter similarity.*

Intuitively, when two clients have similar data distributions, the stochastic clause updates in each TM are likely to reinforce the same clause. This alignment leads to a larger overlap, hence a higher Jaccard similarity. A formal proof is provided in Appendix A.1.

Empirical results (Figure 2) show that the Jaccard similarity of clients' learned parameters, $\mathcal{J}(\text{clients})$, is strongly positively correlated with the true label distribution similarity, $\mathcal{J}(\text{true})$, and strongly negatively correlated with the Wasserstein distance between client and true distributions, $W(\text{true})$. This indicates that data heterogeneity can be reliably inferred from observable TM parameters ($\mathcal{J}(\text{clients})$), motivating similarity-driven clause allocation without accessing metadata.

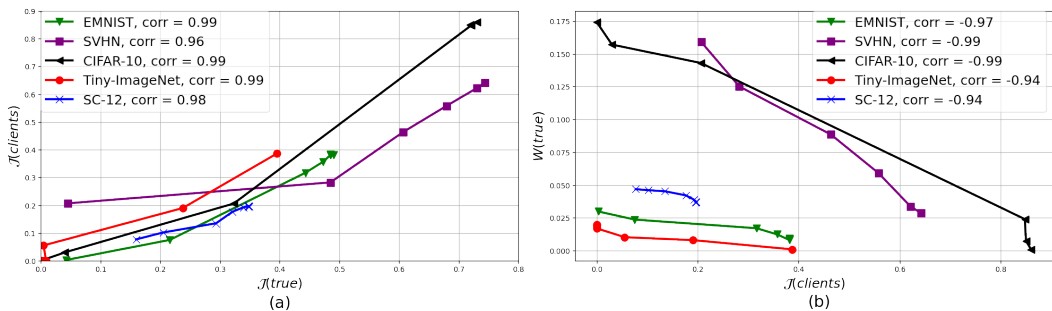

Figure 2: (a) Strong positive correlation and consistent trend between $\mathcal{J}(\text{true})$ and $\mathcal{J}(\text{clients})$. (b) Relationship between $W(\text{true})$ and $\mathcal{J}(\text{clients})$ shows strong negative correlation across datasets.

### 4.3 Algorithm Overview

CS-pFedTM begins with a reference round, in which clients train a tiny reference TM and upload their parameters to the server. These reference parameters serve two key purposes. Firstly, they enable the server to estimate the communication cost per clause and, given the communication budget for downloading the global model, determine the minimum fraction of clauses that must remain local. Secondly, they provide a basis for computing client parameter similarity, which serves as a proxy for data heterogeneity. This similarity-driven measure is then used to set the local-global clause allocation for the system: when the participating clients exhibit higher overall heterogeneity, the scheme emphasizes more local clauses to improve personalization, whereas for lower heterogeneity, more global clauses are used for knowledge sharing. For subsequent rounds, clients are randomly sampled as usual, but only the top-performing clients' states (based on local performance) are uploaded and used in global aggregation. This ensures that the global model incorporates the most informative updates while maintaining fairness in client participation.

Based on the observed parameter similarity and communication budget, the server allocates local and global clauses for each client accordingly. Algorithm 3 summarizes the full approach.

---

**Algorithm 1 CS-pFedTM: Communication-Efficient and Similarity-based Personalized FL with TM**

---

**Input:** Total number of clients $N_c$, total communication rounds $T$, number of clauses per client $n_{\text{clauses}}$, communication budget $\tau$

    **for** round $t = 0, 1, \ldots, T$ **do**

        Server randomly samples $N_t$ clients, $\mathcal{C}_t$

        **if** $t == 0$ **then**

            Clients train a tiny reference TM and upload state parameters

            $\texttt{min\_frac} \leftarrow$ **compute\_min\_frac**

            $JS_{\text{clients}} \leftarrow$ **compute\_client\_similarity**

            $\texttt{local\_frac} \leftarrow \exp\left(-\ln(1/\texttt{min\_frac}) \cdot JS_{\text{clients}}\right)$

            Assign local and global clauses:

$$n_{\text{local}} = \lfloor n_{\text{clauses}} \cdot \texttt{local\_frac} \rfloor, \quad n_{\text{global}} = n_{\text{clauses}} - n_{\text{local}}$$

        **for** each client $n \in \mathcal{C}_t$ **do**

            Client trains local model $L^n$, global model $G^n$

            $L^n, G^n \leftarrow$ **mask\_weights**$(L^n)$, **mask\_weights**$(G^n)$

            Client uploads global parameters $G^n$ to the server

        $G_t \leftarrow$ **aggregate\_global\_models**

        Server updates clients' global TM with $G_t$

    **return** Personalized TMs for each client: $TM^n \in \{G_t, L^n\}$, combined using Equation 3

---

### 4.3.1 COMMUNICATION-AWARE CLAUSE ALLOCATION

To address client heterogeneity under communication constraints, we introduce a communication-aware allocation mechanism. Given a communication budget $\tau$, which specifies the maximum number of megabytes that each client can communicate per round, we first use the reference TM to estimate the per-clause communication footprint, including clause weights and states. This enables us to translate the abstract budget $\tau$ into a concrete bound on the number of clauses that can be shared globally without exceeding this budget.

From this bound, we compute `min_frac`, the minimum fraction of clauses that must remain local. This ensures that each client retains enough locally trained clauses that adhere to the communication budget while still benefiting from global aggregation. By enforcing this budget-driven lower bound, the mechanism prevents infeasible allocations, preserves fairness across heterogeneous clients, and provides a stable foundation for similarity-driven personalization, which dynamically allocates clauses according to data heterogeneity.

### 4.3.2 SIMILARITY-DRIVEN PERSONALIZATION

Within this communication limit, we further adapt clause allocation based on data heterogeneity. As shown in Figure 1, higher heterogeneity ($W(\text{true})$) favors larger local fractions. Since $W(\text{true})$ is unobservable in FL, we approximate it with $\mathcal{J}(\text{clients})$, the average similarity between clients' TM parameters. Empirical results reveal a strong inverse relationship between $\mathcal{J}(\text{clients})$ and $W(\text{true})$: as clients' data distributions diverge further from the true distribution of the system, their parameters become less similar.

We model this in a stable and bounded manner using a decreasing exponential function, which naturally captures the diminishing effect of increasing similarity. When clients are very dissimilar (high heterogeneity), the exponential term is large, resulting in a higher allocation of local clauses, emphasizing personalization. Conversely, as clients become more similar (low heterogeneity), the exponential term decreases rapidly, reducing the local fraction and favoring shared global knowledge. This formulation ensures that even small differences in similarity among highly heterogeneous clients produce meaningful increases in local clause allocation, while clients that are already similar are quickly shifted toward increased global aggregation. Furthermore, by setting:

$$c = \ln(1/\texttt{min\_frac}),$$

we guarantee $\exp(-c \cdot \mathcal{J}(\text{clients})) \geq \texttt{min\_frac}$, ensuring that the allocation never falls below the budget-driven minimum.

The local allocation threshold is therefore defined as:

$$\texttt{local\_frac} = \exp\big(-c \cdot \mathcal{J}(\text{clients})\big)$$

The number of local and global clauses is then computed as

$$n_{local} = \lfloor n_{\text{clauses}} \cdot \texttt{local\_frac} \rfloor, \quad n_{\text{global}} = n_{\text{clauses}} - n_{\text{local}}.$$

Hence, by directly linking clause allocation to the derived similarity measure, CS-pFedTM achieves communication- and heterogeneity-aware personalization.

## 5 EXPERIMENTS

**Benchmark Datasets:** We performed experiments on five image datasets and an audio dataset commonly featured in the FL literature: SVHN (Netzer et al., 2011), EMNIST (Cohen et al., 2017), CIFAR-10, CIFAR-100 (Krizhevsky, 2009), Tiny-ImageNet (Le & Yang, 2015) and SpeechCommands-12 (SC-12) (Warden, 2018).

**Baseline Methods:** To ensure a fair comparison, we evaluated several parameter-decoupling personalization approaches alongside CS-pFedTM. FedAvg serves as the standard FL benchmark (McMahan et al., 2016), while FedAvg++ adds local fine-tuning (Jiang et al., 2023). pFedFDA addresses the bias-variance trade-off via generative classifiers and feature distribution adaptation (Mclaughlin & Su, 2024). FedPAC aligns local and global feature representations using a regularization term (Xu

Table 1: Accuracy (%) of the algorithms for the FL with data heterogeneity and CC - Communication Costs (Upload/Download) for all clients per communication round

| | SVHN | | | | EMNIST | | | | SC-12 | | | | CIFAR-10 | | | | CIFAR-100 | | | | Tiny-ImageNet | | | | Avg. |
|---|---|---|---|---|---|---|---|---|---|---|---|---|---|---|---|---|---|---|---|---|---|---|---|---|---|
| | 0.05 | | 0.1 | | 0.05 | | 0.1 | | 0.05 | | 0.1 | | 0.05 | | 0.1 | | 0.05 | | 0.1 | | 0.05 | | 0.1 | | Acc. |
| | Acc | CC | Acc | CC | Acc | CC | Acc | CC | Acc | CC | Acc | CC | Acc | CC | Acc | CC | Acc | CC | Acc | CC | Acc | CC | Acc | CC | |
| FedAvg | 29.16±14.42 | 13/43 | 53.84±2.71 | 13/43 | 71.41±4.30 | 15/50 | 56.31±2.73 | 15/50 | 56.37±2.93 | 42/141 | 65.46±0.67 | 42/141 | 31.23±0.82 | 13/43 | 32.99±0.46 | 13/43 | 6.58±0.04 | 14/48 | 6.82±0.15 | 14/48 | 1.50±0.09 | 16/53 | 1.48±0.05 | 16/53 | 34.42 |
| FedAvg++ | 80.08±2.21 | " | 71.05±3.50 | " | 72.71±0.85 | " | 74.54±0.32 | " | 67.82±2.72 | " | 66.35±4.03 | " | 79.12±2.75 | " | 67.41±2.33 | " | 43.20±0.26 | " | 34.57±0.91 | " | 19.42±2.32 | " | 14.79±2.40 | " | 57.59 |
| pFedTM | 81.28±2.77 | " | 70.58±3.87 | " | 95.73±0.14 | " | 94.26±0.20 | " | 90.57±0.62 | " | 91.11±1.82 | " | 85.60±1.71 | " | 77.05±0.78 | " | 47.03±1.25 | " | 38.47±1.91 | " | 28.03±0.89 | " | 23.22±0.74 | " | 68.58 |
| FedPAC | 83.03±2.06 | " | 82.96±1.79 | " | **95.77±0.50** | 14/46 | **94.28±0.17** | 14/46 | 86.67±4.27 | " | 90.20±1.45 | " | 85.28±1.37 | " | 79.17±0.75 | " | 45.46±0.60 | " | 37.11±0.70 | " | 28.60±0.36 | 13/43 | 21.59±0.32 | 13/43 | 69.18 |
| FedRep | 80.81±3.13 | " | 81.33±2.53 | " | 78.12±0.17 | " | 78.18±0.32 | " | 88.49±2.46 | " | 82.91±3.10 | " | 86.43±1.45 | " | 79.48±1.23 | " | 44.44±1.56 | " | 38.01±1.53 | " | 27.47±0.84 | " | 20.77±0.57 | " | 65.54 |
| FedPer | 83.27±2.00 | " | 76.12±1.29 | " | 94.37±1.00 | " | 92.68±0.37 | " | 90.82±0.48 | " | 90.97±2.87 | " | 83.76±1.63 | " | 76.13±1.29 | " | 43.02±0.54 | " | 34.66±0.82 | " | 26.27±0.04 | " | 19.89±0.50 | " | 67.66 |
| LG-FedAvg | 84.20±1.96 | 0.67/1.0 | 78.95±1.01 | 0.67/1.0 | 75.80±0.52 | 4.26/4.7 | 75.69±0.08 | 4.26/4.7 | 79.42±2.62 | 0.41/0.62 | 76.72±1.64 | 0.41/0.62 | 84.22±1.60 | 0.67/1.0 | 75.56±0.52 | 0.67/1.0 | 37.88±1.02 | 6.7/10 | 29.07±0.74 | 6.7/10 | 22.62±1.26 | 13/20 | 15.54±0.33 | 13/20 | 61.30 |
| FedSelect (0.3) | 79.51±2.25 | 2.3/2.3 | 67.90±1.54 | 2.3/2.3 | 94.18±0.04 | 2.7/2.7 | 91.51±0.25 | 2.7/2.7 | 91.07±0.19 | 7.5/7.5 | 85.83±0.24 | 7.5/7.5 | 85.95±0.66 | 2.3/2.3 | 78.47±0.68 | 2.3/2.3 | 47.75±1.34 | 2.6/2.6 | 37.13±0.61 | 2.6/2.6 | 29.11±0.60 | 2.9/2.9 | 22.42±1.83 | 2.9/2.9 | 67.57 |
| FedSelect (1.0) | 79.45±1.86 | 7.7/7.7 | 68.88±1.74 | 7.7/7.7 | 94.55±0.48 | 8.9/8.9 | 91.78±0.10 | 8.9/8.9 | 91.16±0.16 | 25/25 | 85.85±0.17 | 25/25 | 86.37±0.72 | 7.7/7.7 | 78.81±0.93 | 7.7/7.7 | 47.76±0.45 | 8.6/8.6 | 36.04±0.13 | 8.6/8.6 | **30.02±0.74** | 9.6/9.6 | 22.45±1.26 | 9.6/9.6 | 67.76 |
| TPFL | 86.64±0.71 | 0.12/2.4 | 80.39±0.88 | 0.12/2.4 | 91.99±0.23 | 0.08/5.8 | 89.05±0.22 | 0.08/4.4 | 83.94±1.40 | 0.19/7.7 | 79.48±3.17 | 0.22/7.7 | 85.10±1.01 | 0.08/16 | 77.97±1.37 | 0.08/16 | 41.72±0.76 | 0.04/5.6 | 31.68±0.48 | 0.04/5.6 | 15.59±0.43 | 0.02/5.2 | 11.41±0.61 | 0.02/4.1 | 64.58 |
| FedTM | 55.58±1.13 | 0.33/12 | 59.02±3.77 | 0.33/12 | 62.94±1.87 | 1.4/48 | 69.44±1.87 | 1.4/48 | 62.33±0.27 | 1.2/35 | 62.37±0.21 | 1.2/35 | 37.86±1.90 | 0.37/15 | 39.62±0.31 | 0.37/15 | 4.37±0.06 | 1.3/46 | 4.52±0.57 | 1.3/46 | 3.67±0.06 | 1.4/53 | 3.43±0.26 | 1.4/53 | 38.76 |
| CS-pFedTM | **89.59±0.78** | 0.01/0.28 | **83.91±1.61** | 0.05/0.96 | 94.60±0.37 | 0.02/0.5 | 91.51±0.54 | 0.11/2.2 | **91.16±2.64** | 0.01/0.35 | **91.76±1.21** | 0.02/0.56 | **86.92±0.83** | 0.03/0.76 | **79.81±0.68** | 0.03/0.99 | **48.20±0.85** | 0.04/0.88 | **39.03±0.69** | 0.04/0.88 | 29.25±0.66 | 0.12/2.6 | **24.20±0.09** | 0.12/2.6 | **70.82** |

et al., 2023). FedRep and FedPer communicate only base layers, retraining classifier heads or the full model for personalization (Collins et al., 2023; Arivazhagan et al., 2019). LG-FedAvg transmits only the global classifier and linearly combines local and global layers (Liang et al., 2020). FedSelect personalizes subnetworks via selective masking but limits aggregation to participating clients, leaving non-participating clients without updates (Tamirisa et al., 2024). We also include TM-based FL methods. FedTM, which performs sample-based aggregation (How et al., 2023) and TPFL that addresses heterogeneity via confidence-based clustering (Gohari et al., 2024).

**FL Configuration:** Following standard practice (Hsu et al., 2019; Jiang et al., 2023; Mclaughlin & Su, 2024), we simulate heterogeneity using a Dirichlet partition with $\alpha \in \{0.1, 0.05\}$ and a 0.3 participation rate over 100 clients. Clients train for 1 local epoch per round, and results report the best average personalized accuracy over 100 rounds (3 seeds). Following the original FedTM paper (How et al., 2023), we train FedTM with 5 local epochs as its non-personalized aggregation requires multiple steps to produce stable updates. Communication cost is measured as total uploaded/downloaded parameters per round. FedSelect is adapted to the cross-device setting with 0.3 client participation, with full participation also reported for consistency as in Tamirisa et al. (2024).

**Model Configuration:** We used a 2-layer CNN (Xu et al., 2023) for the image datasets and the CNN from Zhang et al. (2018) for SC-12, trained with batch size 128 (Liang et al., 2020). FedTM and CS-pFedTM use CTMs, while TPFL uses a Coalesced TM configured with the same total number of clauses for fairness. CS-pFedTM's download budget $\tau$ is set to match the most download-efficient baseline, ensuring comparable communication conditions.

## 5.1 PERFORMANCE

CS-pFedTM achieves accuracy comparable to state-of-the-art personalized FL methods and delivers the highest average performance across all heterogeneous settings (Table 1). It outperforms the second-best method by an average of 1.64%, and surpasses TM-based FL FedTM and TPFL in all settings by an average of 32.1% and 6.24% respectively. We also benchmarked CS-pFedTM against sparsification-based personalization methods. As DisPFL is decentralized and SpaFL requires larger CNNs for pruning, we report these results in Appendix C.1, where CS-pFedTM maintains superior performance and efficiency.

## 5.2 COMMUNICATION COSTS

Communication costs are critical in FL, especially for edge devices with limited bandwidth (Asad et al., 2023). Table 1 shows that CS-pFedTM achieves the lowest overall communication costs among all evaluated methods. This reduction is primarily due to CS-pFedTM's design, which uploads only global parameters guided by client heterogeneity and communication budgets, rather than the full model, while the bit-based CTM representation additionally reduces memory requirements compared to full-precision CNNs (Lei et al., 2020). As a result, CS-pFedTM achieves $31.3\times$ and $45.8\times$ lower upload and download costs than FedTM. On average, CS-pFedTM is $85.8\times$ more upload-efficient and $5.58\times$ more download-efficient compared to LG-FedAvg, and $158\times$ and $6\times$ more efficient compared to FedSelect, while delivering superior model performance. FedTM demonstrates lower upload costs compared to LG-FedAvg, yet remains less efficient in download costs. Although FedPAC surpasses CS-pFedTM in terms of accuracy on the EMNIST dataset, CS-pFedTM remains an average of $876\times$ more upload-efficient and $106\times$ more download-efficient. Furthermore, CS-pFedTM reduces upload and download communication by $3.6\times$ and $9.28\times$ compared to TPFL. These results show that CS-pFedTM offers the most communication-efficient solution, making it ideal for bandwidth-limited FL.

## 5.3 MEMORY COSTS AND TRAINING LATENCY

Table 2: Average Memory Storage (MS) and Runtime Memory (RTM) in MB and Training Latency (L) in seconds on each client

| | SVHN | | | EMNIST | | | CIFAR-10 | | | CIFAR-100 | | | SC-12 | | | Tiny-ImageNet | | |
|---|---|---|---|---|---|---|---|---|---|---|---|---|---|---|---|---|---|---|
| | MS | RTM | L | MS | RTM | L | MS | RTM | L | MS | RTM | L | MS | RTM | L | MS | RTM | L |
| CNN | 0.43 | 101 | 1.48 | 0.50 | 50.6 | 3.82 | 0.43 | 111 | 1.52 | 0.48 | 118 | 1.47 | 1.41 | 278 | 1.25 | 0.53 | 144 | 3.23 |
| CoTM | **0.02** | 22.5 | 2.92 | **0.06** | 71.5 | 17.1 | **0.06** | 41.6 | 2.44 | **0.05** | 28.2 | 3.36 | **0.06** | 8.6 | 5.37 | **0.05** | 57.3 | 10.6 |
| CTM | 0.12 | **22.1** | **0.70** | 0.48 | **47.6** | 3.50 | 0.15 | **31.8** | **0.63** | 0.46 | **25.7** | **1.24** | 0.35 | **10.8** | **0.78** | 0.53 | **42.5** | **2.45** |

We evaluated runtime memory and model storage for all TM variants. As shown in Table 2, CTMs are significantly more efficient than CNNs, requiring $2.17\times$ less storage and $7.16\times$ lower runtime memory. TPFL's CoTM further reduces static model size by $7.76\times$, but this comes at the cost of $1.17\times$ higher runtime memory and $4.48\times$ higher latency. More importantly, at the FL system level, CS-pFedTM achieves $6.24\%$ higher accuracy while reducing upload and download communication by $3.6\times$ and $9.28\times$. Since communication and accuracy, not static model size, are the dominant constraints in practical FL deployments (Khan et al., 2021), CS-pFedTM provides a strictly better performance, communication and runtime. Despite TPFL's smaller model footprint, CS-pFedTM is therefore more suitable for resource-constrained, bandwidth-limited FL environments.

## 5.4 EFFECT OF HETEROGENEITY

To analyze heterogeneity, we varied the number of classes per client in CIFAR-10, with fewer classes indicating higher heterogeneity. From Figure 3, CS-pFedTM achieves the largest gains under highly heterogeneous settings, though its advantage slightly decreases as heterogeneity lowers. It consistently outperforms communication-efficient baselines such as LG-FedAvg, TPFL and FedSelect. Like CNN-based methods, stronger performance under lower heterogeneity often requires more shared global parameters, a trend CS-pFedTM follows. For the higher budget setting, we constrained CS-pFedTM's communication to the maximum used by competing methods; even so, it incurs significantly lower costs while closing the performance gap. Another factor partly explaining this gap is that TMs are generally less robust than CNNs; however, CS-pFedTM remains the strongest TM-based FL method across all heterogeneity levels, and recent advances in TM architectures such as GraphTM indicate promising directions for further performance improvement (Granmo et al., 2025).

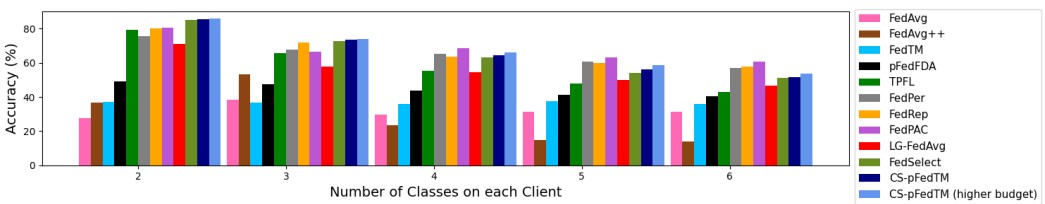

Figure 3: Performance of the algorithms on varying heterogeneity

## 6 CONCLUSIONS

We presented CS-pFedTM, an efficient personalized FL framework with TMs that jointly leverages local and global models through a similarity-based clause allocation mechanism that adapts to heterogeneity and communication constraints. CS-pFedTM achieves substantial resource reductions, at least $3.6\times$ in upload, $5.58\times$ in download, $1.17\times$ in runtime memory, and $1.62\times$ in training latency, without compromising accuracy. By focusing on clause-level optimization, this work lays the groundwork for future improvements such as weight optimization, adaptive mask learning, and clause sparsification. Additionally, the observed link between parameter similarity and data distribution provides insights for FL extensions, including resource-aware personalization and dynamic clause adaptation to handle concept drift.

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

# A APPENDIX

## A.1 PROOF OF RELATION BETWEEN WASSERSTEIN DISTANCE AND JACCARD SIMILARITY

Training in a Tsetlin Machine (TM) is stochastic because of:

- Random selection of clauses for updating, and
- Randomized rewards and penalties from Type I feedback.

This stochasticity makes the learned clause states highly sensitive to the underlying data distribution. Motivated by this, we investigate how differences in data distributions across clients—quantified via the Wasserstein distance—affect the similarity of their learned parameters, measured using Jaccard similarity.

**Definition 1 (1-Wasserstein Distance (Kolouri et al., 2017))** *The 1-Wasserstein distance between two distributions $q_1$ and $q_2$ over a metric space $Z$ is*

$$W(q_1, q_2) := \inf_{q \in Q(q_1, q_2)} \int_{Z \times Z} d(z_1, z_2) \, dq(z_1, z_2),$$

*where $d(\cdot, \cdot)$ is a distance function and $Q(q_1, q_2)$ denotes the set of couplings with marginals $q_1$ and $q_2$.*

**Lemma 2 (Distributional Dissimilarity)** *Let $q_1, q_2, q_2'$ be data distributions. If $W(q_1, q_2') > W(q_1, q_2)$, then $q_2'$ is more dissimilar to $q_1$ than $q_2$ is.*

By the definition of the 1-Wasserstein distance and the principle of optimal transport (Kolouri et al., 2017), a larger value indicates that, on average, it is "harder" to transport samples from $q_1$ to $q_2$. Hence, if $W(q_1, q_2') > W(q_1, q_2)$, the distribution $q_2'$ is more dissimilar to $q_1$ than $q_2$ is. Intuitively, samples from $q_2'$ are less likely to resemble samples from $q_1$ compared to samples from $q_2$.

Next, we define the similarity of parameters by

**Definition 2 (Jaccard Similarity (Costa, 2021))** *The Jaccard similarity between two sets of binary vectors $S_A$ and $S_B$ is the size of the intersection divided by the size of the union of the sets:*

$$\mathcal{J}(S_A, S_B) = \frac{|S_A \cap S_B|}{|S_A \cup S_B|}.$$

To compare the states between two sets of clauses $A$ and $B$:

- $|S_A \cap S_B|$: represents the number of clauses that are active in both sets, (clauses that include at least one literal in both)
- $|S_A \cup S_B|$: represents the number of clauses that contain literals in either $S_A$ or $S_B$.

The Jaccard similarity between two states, $S_A$ and $S_B$, therefore measures the degree of overlap in active clauses between the two states. Higher values indicate that the same clauses has at least an include action in both states, reflecting the similarity in how feedback has shaped the clauses during training.

**Corollary 3 (Inverse Relation Between Distribution Divergence and Clause Overlap)** *Let $q_A$ and $q_B$ be two class distributions and $S_A$, $S_B$ be the corresponding trained TM states (sets of clauses). Then:*

$$W(q_A, q_B) \longrightarrow \text{smaller} \implies \mathcal{J}(S_A, S_B) \longrightarrow \text{larger},$$

*Thus, lower distributional divergence corresponds to higher parameter similarity.*

Training is done one sample at a time. Given a Multi-Class TM with $M$ classes, when training an input $\mathbf{x}$ from class $y = k$, the TA teams associated with class $k$ will be trained to output $y = 1$ and the other classes' ($y \neq k$) TA teams will be selected to train to output $y = 0$ on the training input

**x**. The probability of selecting a TA team from class $m \in \{1, ..., M\}$ for positive training can be defined as $q_m$ where $\sum_{m=1}^{M} q_m = 1$.

Let $c_k^j$ be the $j$-th clause for class $k$, with $C$ total clauses per class. Let $L_k^j$ denote the number of literals included in the TA teams of $c_k^j$. During training of $(\mathbf{x}, y)$ with class label $y$:

- Type I feedback reinforces the TA teams of clauses corresponding to $y = k$ are reinforced to include literals matching the input, increasing the likelihood that the clause outputs 1: $\Delta L_k^j \mid y = k \geq 0$.

- Type II feedback guides the TAs in clauses of other classes $y \neq k$ to include zero-valued literals or suppress active literals. This reduces the likelihood of false positives, effectively decreasing the number of literals contributing to the clause output: $\Delta L_k^j \mid y \neq k \leq 0$.

We define the change in the number of literals included in clause $c_k^j$ as $\Delta L_k^j$, which depends on Equation 2 and the specific training sample. Let

$$\delta_+ := \mathbb{E}[\Delta L_k^j \mid y = k] \geq 0, \qquad \delta_- := \mathbb{E}[-\Delta L_k^j \mid y \neq k] \geq 0,$$

represent the expected increase in literals for Type I feedback and the expected decrease in literals for Type II feedback, respectively.

Then, for a data distribution $q = \{q_k\}_{m=1}^{M}$, the expected number of literals in clause $c_k^j$ after training is

$$\mathbb{E}[L_k^j \mid q_k] = q_k \, \delta_+ + (1 - q_k) \, \delta_- \tag{4}$$
$$= \delta_- + q_k \, (\delta_+ - \delta_-), \tag{5}$$

This expression captures the average effect of Type I and Type II feedback across the class distribution.

Let two datasets, $A$ and $B$ have distributions $q^A = \{q_k^A\}_{k=1}^{M}$ and $q^B = \{q_k^B\}_{k=1}^{M}$,

$$S_A = \{c_k^j : L_k^{j,A} \geq 1, \forall k \in M, j \in C\}, \qquad S_B = \{c_k^j : L_k^{j,B} \geq 1, \forall k \in M, j \in C\},$$

denote the sets of clauses that contain at least one include action, respectively.

We define indicator variables for literals present in clauses:

$$I_k^{j,X} := \mathbf{1}\{L_k^{j,X} \geq 1\}, \quad X \in \{A, B\}.$$

Thus, a clause $c_k^j$ belongs to $S_X$ if and only if $I_k^{j,X} = 1$.

The size of the overlap between the two sets is the dot product of the indicator vectors:

$$|S_A \cap S_B| = \sum_{k=1}^{M} \sum_{j=1}^{C} I_k^{j,A} \cdot I_k^{j,B}.$$

Taking expectations, we obtain

$$\mathbb{E}[|S_A \cap S_B|] = \sum_{k=1}^{M} \sum_{j=1}^{C} \mathbb{E}[I_k^{j,A} I_k^{j,B}].$$

Since the $\text{TM}_A$ and $\text{TM}_B$ trained on $q^A$ and $q^B$ are independent, the expectation can be expressed as:

$$\mathbb{E}[|S_A \cap S_B|] = \sum_{k=1}^{M} \sum_{j=1}^{C} \mathbb{E}[I_k^{j,A}] \, \mathbb{E}[I_k^{j,B}].$$

Since $\mathbb{E}[I_k^{j,X}] = \Pr(I_k^{j,X} = 1) = \Pr(L_k^{j,X} \geq 1)$, we get:

$$\mathbb{E}[|S_A \cap S_B|] = \sum_{k=1}^{M} \sum_{j=1}^{C} \Pr(L_k^{j,A} \geq 1) \cdot \Pr(L_k^{j,B} \geq 1).$$

Approximating them using normalized expected literal counts:

For the bounded random variable $L_k^{j,X} \in [0, L_{\max}]$, we can bound $\Pr(L_k^{j,X} \geq 1)$ using its expectation:

$$\mathbb{E}[L_k^{j,X}] = \mathbb{E}[L_k^{j,X} \mid L_k^{j,X} \geq 1] \Pr(L_k^{j,X} \geq 1) + \mathbb{E}[L_k^{j,X} \mid L_k^{j,X} < 1] \Pr(L_k^{j,X} < 1).$$

Since $0 \leq L_k^{j,X} \leq L_{\max}$ on the event $\{L_k^{j,X} \geq 1\}$, we have

$$\Pr(L_k^{j,X} \geq 1) \leq \mathbb{E}[L_k^{j,X}],$$

and similarly

$$\mathbb{E}[L_k^{j,X}] \leq L_{\max} \Pr(L_k^{j,X} \geq 1),$$

which implies

$$\frac{\mathbb{E}[L_k^{j,X}]}{L_{\max}} \leq \Pr(L_k^{j,X} \geq 1) \leq \mathbb{E}[L_k^{j,X}].$$

Since the distribution of $L_k^{j,X}$ is typically spread across $[0, L_{\max}]$, its normalized expectation provides a tractable approximation:

$$\Pr(L_k^{j,X} \geq 1) \approx \frac{\mathbb{E}[L_k^{j,X} \mid q_k^X]}{L_{\max}}.$$

Substituting this approximation, the expected overlap becomes

$$\mathbb{E}[|S_A \cap S_B|] \approx \sum_{k=1}^{M} \sum_{j=1}^{C} \frac{\mathbb{E}[L_k^j | q_k^A]}{L_{\max}} \cdot \frac{\mathbb{E}[L_k^j | q_k^B]}{L_{\max}}$$

**Definition 3 (Arithmetic Mean-Geometric Mean Inequality (Xia et al., 1999))** *For non-negative numbers $a_1, a_2, \ldots, a_M$,*

$$\frac{a_1 + a_2 + \cdots + a_M}{M} \geq \sqrt[M]{a_1 a_2 \cdots a_M},$$

*with equality if and only if $a_1 = a_2 = \cdots = a_M$.*

Applying the AM–GM inequality gives

$$\frac{\mathbb{E}[L_k^j \mid q_k^A]}{L_{\max}} \cdot \frac{\mathbb{E}[L_k^j \mid q_k^B]}{L_{\max}} \leq \left( \frac{\frac{\mathbb{E}[L_k^j | q_k^A]}{L_{\max}} + \frac{\mathbb{E}[L_k^j | q_k^B]}{L_{\max}}}{2} \right)^2.$$

Hence, each product term is maximized when

$$\mathbb{E}[L_k^j \mid q_k^A] = \mathbb{E}[L_k^j \mid q_k^B].$$

Therefore, summing over all classes and clauses, the total expected overlap is maximized when the two data distributions are aligned:

$$q_k^A = q_k^B \quad \forall k \in M.$$

Therefore, the expected clause overlap $\mathbb{E}[|S_A \cap S_B|]$ is maximized when the class distributions are identical $W(q^A, q^B) \to 0$. Smaller distributional distance between $q^A$ and $q^B$ implies higher expected Jaccard similarity of the clause-activity states; conversely, larger distributional divergence generally reduces clause overlap.

## B EXPERIMENTAL DETAILS

### B.1 DATASETS

We evaluated the different approaches on the SVHN (Netzer et al., 2011), Extended MNIST (EM-NIST) (Cohen et al., 2017), CIFAR-10 , CIFAR-100 (Krizhevsky, 2009), SpeechCommands (Warden, 2018) dataset and Tiny-ImageNet (Le & Yang, 2015). All datasets are downloaded and preprocessed with PyTorch (Paszke et al., 2019).

- SVHN: This dataset is imbalanced and consists of digits and numbers captured in natural scenes, presenting a more challenging real-world problem (Netzer et al., 2011).
- EMNIST: The extended version of MNIST which contains 814,255 characters with 62 unbalanced classes. Similar to BiFL (Yang et al., 2021; Marfoq et al., 2022), we only used a subset of the entire dataset for training and testing.
- CIFAR-10: A real-world image dataset of 10 classes with 6000 images per class (Krizhevsky, 2009).
- CIFAR-100: A real-world image dataset of 100 classes with 6000 images per class (Krizhevsky, 2009).
- SpeechCommands-12 (SC-12): A dataset containing 10 spoken keywords ('Yes', 'No', 'Left', 'Right', 'Up', 'Down', 'Stop', 'Go', 'On', 'Off') with the remaining 20 keywords labelled as 'silence' and 'unknown' (Warden, 2018).
- Tiny-ImageNet: A dataset containing 100000 real-world images of 200 classes, downsized to 64×64 colored images (Le & Yang, 2015).

For the SpeechCommands-12 dataset, we preprocessed each audio clip and extracted 40x49 MFCC features as defined in (Zhang et al., 2018) for the DNN-based algorithms while we extracted 13x29 MFCC features as defined in (Lei et al., 2021) for the TM-based algorithms.

### B.2 LIBARIES AND MACHINE

To evaluate the average run-time memory usage and training latency, these were estimated by containerizing the PyPi memory-profiler package in Docker using 2 CPUs.

#### B.2.1 BASELINE MODELS CONFIGURATION

In configuring all baseline models, we performed parameter tuning to optimize their performance. specifically, for the learning rate if not defined in the original paper, we explored these values: [0.01, 0.05, 0.1].

### B.3 CS-PFEDTM MODEL CONFIGURATION

To meet the booleanized input requirements essential for TMs, we implemented distinct pre-processing steps for each of our datasets. For the EMNIST dataset, we encoded the data by setting pixel values larger than 40 to 1, and values below or equal to 40 to 0. For the SVHN dataset, we binarized the data using an adaptive Gaussian thresholding procedure with a window size of 11 and a threshold value of 2 (Granmo et al., 2019). For the CIFAR-10 dataset, we booleanized using 3x3 color thermometer encoding and for the CIFAR-100 and Tiny-ImageNet, we booleanized using 2x2 color thermometer encoding(Granmo, 2023). Across all datasets, we utilized the CTM, adjusting parameters such as the number of clauses, feedback threshold, learning sensitivity, and patch dimension. We set $\delta = 0.5$ for **AverageCW** to average the local weights.

## C ADDITIONAL RESULTS

### C.1 COMPARISON WITH SPARSIFICATION METHODS

Sparsification can also be leveraged as a form of personalization by selectively pruning model components based on their importance to each client. We compare our method with DisPFL (Dai et al.,

Table 3: CS-pFedTM model configuration

|  |  | SVHN | EMNIST | CIFAR-10 | CIFAR-100 | SC-12 | Tiny-ImageNet |
|---|---|---|---|---|---|---|---|
| Dir(0.05) | Local Clauses | 293 | 193 | 190 | 103 | 792 | 57 |
|  | Global Clauses | 7 | 2 | 10 | 2 | 8 | 3 |
| Dir(0.1) | Local Clauses | 276 | 186 | 187 | 103 | 787 | 57 |
|  | Global Clauses | 24 | 9 | 13 | 2 | 13 | 3 |
|  | Feedback Threshold | 500 | 100 | 150 | 1000 | 200 | 2000 |
|  | Learning Sensitivity | 7.5 | 5 | 5 | 5 | 5 | 1.5 |
|  | Patch Dimensions | (5,5) | (10,10) | (3,3) | (2,2) | (10,10) | (2,2) |

2022) and SpaFL Kim et al. (2024), two communication-efficient personalized FL approaches that uses sparsification.

Since DisPFL is a decentralized FL method, we focus on the average per-round communication cost per client when sharing parameters with neighbors, and compare it with the per-client communication cost of our approach. We utilized the same CNN models as defined in Section 5.

Table 4: Performance of DisPFL (n), where n is the number of neighbours vs CS-pFedTM and CC - Average CC per client per round for SVHN, EMNIST, and SC-12

|  | SVHN | | | | EMNIST | | | | SC-12 | | | |
|---|---|---|---|---|---|---|---|---|---|---|---|---|
|  | $Dir(0.05)$ | | $Dir(0.1)$ | | $Dir(0.05)$ | | $Dir(0.1)$ | | $Dir(0.05)$ | | $Dir(0.1)$ | |
|  | Acc | CC | Acc | CC | Acc | CC | Acc | CC | Acc | CC | Acc | CC |
| DisPFL (n=30) | 77.08±2.17 | 6.46 | 65.09±2.24 | 6.46 | 90.90±0.32 | 7.43 | 88.44±0.41 | 7.43 | 84.34±0.22 | 21.2 | 76.54±0.98 | 21.2 |
| DisPFL (n=10) | 75.96±1.93 | 2.15 | 60.18±1.92 | 2.15 | 90.15±0.38 | 2.48 | 88.89±0.42 | 2.48 | 82.10±0.18 | 7.05 | 76.54±0.35 | 7.05 |
| DisPFL (n=5) | 76.35±2.01 | 1.08 | 61.36±1.85 | 1.08 | 91.15±0.42 | 1.24 | 89.90±0.45 | 1.24 | 81.91±0.11 | 2.38 | 72.30±0.28 | 2.38 |
| CS-pFedTM | **89.59±0.78** | **0.01** | **83.91±1.61** | **0.05** | **94.60±0.37** | **0.02** | **91.51±0.54** | **0.11** | **91.16±2.64** | **0.01** | **91.76±1.21** | **0.02** |

Table 5: Performance of DisPFL (n), where n is the number of neighbours vs CS-pFedTM and CC - Average CC per client per round for CIFAR-10, CIFAR-100, and Tiny-ImageNet

|  | CIFAR-10 | | | | CIFAR-100 | | | | Tiny-ImageNet | | | |
|---|---|---|---|---|---|---|---|---|---|---|---|---|
|  | $Dir(0.05)$ | | $Dir(0.1)$ | | $Dir(0.05)$ | | $Dir(0.1)$ | | $Dir(0.05)$ | | $Dir(0.1)$ | |
|  | Acc | CC | Acc | CC | Acc | CC | Acc | CC | Acc | CC | Acc | CC |
| DisPFL (n=30) | 82.26±1.12 | 6.5 | 74.57±1.04 | 6.5 | 30.52±1.30 | 7.14 | 23.38±0.42 | 7.14 | 22.55±0.43 | 7.93 | 17.09±0.62 | 7.93 |
| DisPFL (n=10) | 82.10±0.93 | 2.17 | 71.94±1.01 | 2.17 | 30.10±0.82 | 2.38 | 23.22±0.79 | 2.38 | 21.03±0.66 | 2.64 | 15.69±0.59 | 2.64 |
| DisPFL (n=5) | 81.41±0.93 | 1.08 | 73.15±0.94 | 1.41 | 30.46±0.79 | 1.19 | 21.82±0.33 | 1.19 | 19.10±0.46 | 1.32 | 14.79±0.47 | 1.32 |
| CS-pFedTM | **86.92±0.83** | **0.03** | **79.81±0.68** | **0.3** | **48.20±0.85** | **0.04** | **39.03±0.69** | **0.04** | **29.25±0.66** | **0.1** | **24.20±0.09** | **0.1** |

Although decentralized FL methods avoid the server communication bottleneck, they become more communication-intensive when $n > 1$ since each client must exchange updates with multiple neighbors per round. In contrast, centralized FL requires only one upload and one download per client. Our results show that CS-pFedTM consistently outperforms DisPFL across all settings, while also achieving significantly lower per-round communication costs. Nevertheless, one advantage of DisPFL is that the number of neighbors can be predefined, offering flexibility in network topology design. However, this comes at the expense of a trade-off as seen in Table 4 and Table 5, where increasing the number of neighbors may improve information mixing but could lead to higher communication overhead and potentially affect model performance.

To further reduce communication costs beyond parameter or gradient exchange, pruning-based methods have been proposed. In SpaFL, trainable thresholds are assigned to each filter or neuron, which prune their connected parameters to induce structured sparsity. To minimize communication, only these thresholds are exchanged between clients and the server, reducing costs by up to two orders of magnitude compared to transmitting full model parameters Kim et al. (2024).

However, pruning is largely ineffective for smaller CNNs, since their limited parameter counts leave little redundancy to exploit. Therefore, because the CNNs used in Section 5 are too small for pruning, we adopt the larger model from the original SpaFL paper for comparison. Moreover, since SpaFL communicates only thresholds, we evaluate our CS-pFedTM under stricter communication budgets to ensure fairness. The results in Table 6 demonstrate that our method achieves stronger personalization while operating under tighter resource constraints.

Table 6: Comparison of SpaFL and CS-pFedTM: Performance, Communication Costs per client per round, and Model Size after pruning for SpaFL

| | FMNIST | | | | | | CIFAR-10 | | | | | | CIFAR-100 | | | | | |
| | Dir(0.05) | | | Dir(0.1) | | | Dir(0.05) | | | Dir(0.1) | | | Dir(0.05) | | | Dir(0.1) | | |
| | Acc | CC | Size | Acc | CC | Size | Acc | CC | Size | Acc | CC | Size | Acc | CC | Size | Acc | CC | Size |
| SpaFL | 96.72±0.31 | 0.07/0.23 | 0.94 | 95.24±0.42 | 0.07/0.23 | 0.94 | 83.33±0.79 | 0.09/0.26 | 3.23 | 75.57±0.65 | 0.09/0.26 | 3.23 | 45.15±0.94 | 0.29/0.96 | 11.2 | 36.25±0.64 | 0.29/0.96 | 11.2 |
| CS-pFedTM | **97.83±0.44** | **0.02/0.22** | **0.26** | **95.58±0.38** | **0.02/0.24** | **0.26** | **85.34±0.74** | **0.004/0.09** | **0.35** | **78.57±0.69** | **0.06/0.15** | **0.35** | **48.03±0.81** | **0.25/0.85** | **0.83** | **38.16±0.52** | **0.06/1.5** | **0.83** |

## C.2 EFFECT OF PARTICIPATION RATIO AND NUMBER OF CLIENTS

We analyzed the scalability of CS-pFedTM by varying the number of clients from 20 to 500 and adjusting the client participation ratio per communication round to [0.1, 0.3, 0.5, 1.0] on the CIFAR-10 dataset. The results demonstrate that CS-pFedTM consistently delivers strong performance across all configurations, regardless of the total number of clients or the participation rate per round. This shows CS-pFedTM's scalability and robustness, making it well-suited for various FL scenarios.

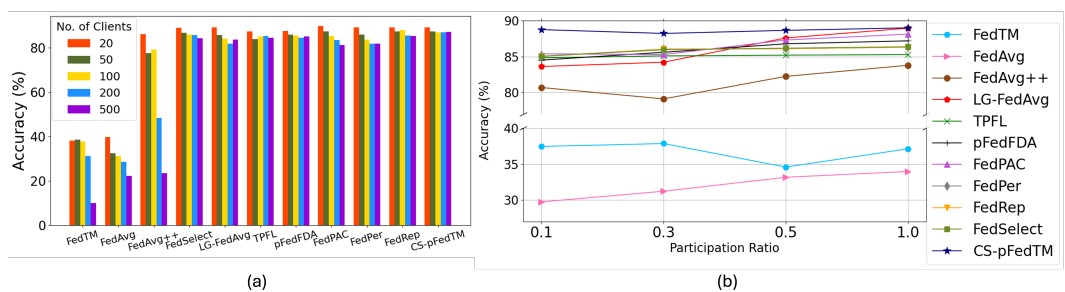

Figure 4: Performance of the algorithms for varying (a) number of clients and (b) participation ratio

## C.3 PERFORMANCE IN EXTREME NON-IID SCENARIOS

Furthermore, it is expected that parameter similarity may lose granularity under extreme non-IID conditions (eg. when clients have completely disjoint label spaces). In such cases, the Jaccard similarity between client clauses can saturate near zero, and the resulting clause allocation becomes highly personalized. This behavior is expected: when clients share almost no structure, the model should tend toward fully personalized learning.

To verify robustness in these extreme regimes, we evaluated CS-pFedTM under Dirichlet settings of $\alpha = 0.01$ and $\alpha = 0.005$. As shown in Table 7, CS-pFedTM continues to match or exceed the performance of all baselines, even as global sharing naturally diminishes. These results indicate that the similarity-driven allocation mechanism remains stable and effective, even when the model transitions toward near-fully personalized operation.

Table 7: Performance of the algorithms in the extreme non-IID setting

| | Dir(0.005) | Dir(0.01) |
| --- | --- | --- |
| FedAvg | 28.18±0.74 | 25.36±0.58 |
| FedAvg++ | 95.91±1.12 | 92.27±0.84 |
| pfedFDA | 94.89±0.66 | 91.86±0.91 |
| FedPAC | 97.81±0.93 | **96.98±0.77** |
| FedRep | 97.25±0.85 | 95.20±0.64 |
| FedPer | 98.04±1.02 | 96.67±0.72 |
| LG-FedAvg | 98.57±0.81 | 94.36±1.10 |
| FedSelect(0.3) | 98.12±0.69 | 96.20±0.95 |
| FedSelect(1.0) | 96.49±0.54 | 95.88±1.16 |
| TPFL | 98.38±1.07 | 96.29±0.91 |
| FedTM | 16.78±0.48 | 15.93±0.62 |
| CS-pFedTM | **98.79±0.78** | 96.72±0.65 |

### C.4 COMPARISON WITH LARGER BASELINE MODEL

Our primary focus is efficiency-oriented personalized FL, where comparisons are typically made under realistic communication and computational constraints. In this setting, lightweight CNNs remain the standard choice across recent personalization literature, as they better reflect practical FL deployments. Nonetheless, we also evaluated the DNN-based FL methods using MobileNet-v2 (Sandler et al., 2019) on CIFAR-10. The outcomes align with the expected behavior of parameter-decoupled FL methods where methods that personalize a substantial portion of the model (eg. LG-FedAvg) retain reasonable performance even with MobileNet, since a large number of parameters are adapted locally. However, methods that personalize only the classifier head (FedPer, FedRep, FedPAC) perform worse than with a 2-layer CNN. This is expected: MobileNet's large shared backbone dominates the representation, and personalizing only the final layer is insufficient to overcome strong distribution shifts under heterogeneous data.

Importantly, even with this much larger backbone model, the communication cost of these MobileNet-based baselines remains several orders of magnitude higher than CS-pFedTM. Despite using a lightweight architecture, CS-pFedTM achieves comparable accuracy while maintaining its primary advantage of reduced communication.

Table 8: Performance of the algorithms with larger models

|  | Dir(0.05) | | Dir(0.1) | |
|---|---|---|---|---|
|  | Acc | CC | Acc | CC |
| FedAvg | 39.73±1.99 | 268/895 | 33.25±1.34 | 268/895 |
| FedAvg++ | 72.4±1.31 | " | 60.53±1.02 | " |
| pfedFDA | 88.01±1.17 | " | 80.93±0.91 | " |
| FedPAC | 75.29±0.96 | 267/890 | 69.64±0.83 | 267/890 |
| FedRep | 80.72±1.25 | " | 78.44±1.11 | " |
| FedPer | 76.79±0.92 | " | 66.17±0.73 | " |
| LG-FedAvg | 87.68±1.83 | 6.66/10.2 | **83.21±1.55** | 6.66/10.2 |
| FedSelect(0.3) | **88.77±0.53** | 7.63/7.63 | 78.76±0.83 | 7.63/7.63 |
| FedSelect(1.0) | 85.71±0.19 | 25.4/25.4 | 80.18±0.22 | 25.4/25.4 |
| CS-pFedTM | 87.34±0.51 | **0.02/0.44** | 80.34±0.96 | **0.26/5.76** |

### C.5 SENSITIVITY ANALYSIS

As shown in Figure 5, our similarity-driven allocation selects the optimal point on each curve, adaptively adjusting the local/global split based on client heterogeneity. Upload communication costs increase as heterogeneity decreases, since more homogeneous clients share a larger fraction of global clauses. These results highlight the trade-off between personalization and communication, demonstrating that our allocation mechanism consistently identifies the best balance across heterogeneity levels.

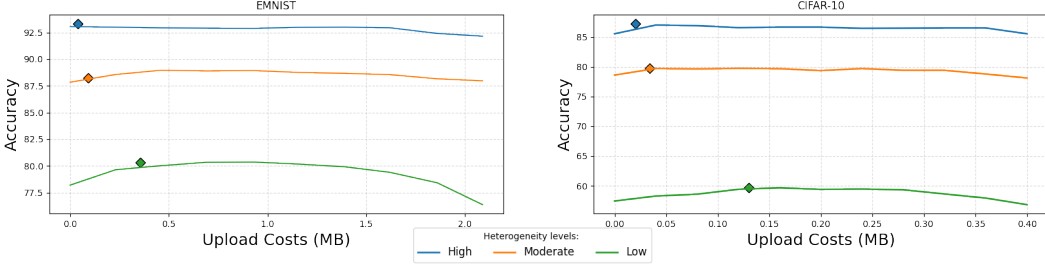

Figure 5: Performance as a function of local clause fraction under different heterogeneity levels. The points indicate the local/global split selected by our similarity-driven allocation, which achieves the highest performance on each curve.

### C.6 Ablation Studies

We conduct ablation experiments to evaluate the individual contributions of CS-pFedTM's two core components: masking and similarity-based personalization. When applied separately, each leads to partial improvements under data heterogeneity. However, the joint application of masking and personalization delivers the best performance, demonstrating that CS-pFedTM consistently achieves the highest accuracy and stability across all datasets.

Table 9: Ablation Studies for CS-pFedTM

|  |  | FedTM | Mask only | Personalization only | CS-pFedTM |
|---|---|---|---|---|---|
| $Dir(0.05)$ | SVHN | 55.58±1.13 | 57.63±2.44 | 87.48±0.94 | **89.59±0.78** |
|  | EMNIST | 62.94±1.87 | 63.57±2.19 | 91.70±1.38 | **94.60±0.37** |
|  | SC-12 | 62.33±0.27 | 63.37±0.87 | 88.82±1.93 | **91.16±2.64** |
|  | CIFAR-10 | 37.86±1.90 | 39.48±0.82 | 85.31±0.91 | **86.92±0.83** |
|  | CIFAR-100 | 4.37±0.06 | 6.61±0.19 | 43.53±0.71 | **48.20±0.85** |
|  | Tiny-Imagenet | 3.67±0.06 | 4.58±0.09 | 21.28±0.58 | **29.25±0.66** |
| $Dir(0.1)$ | SVHN | 59.02±3.77 | 59.66±0.84 | 81.83±0.92 | **83.91±1.61** |
|  | EMNIST | 69.44±1.87 | 71.15±0.42 | 88.49±1.30 | **91.51±0.54** |
|  | SC-12 | 62.37±0.21 | 63.14±0.84 | 91.01±0.07 | **91.76±1.21** |
|  | CIFAR-10 | 39.62±0.31 | 41.98±0.78 | 77.15±0.24 | **79.81±0.68** |
|  | CIFAR-100 | 4.52±0.57 | 9.12±0.49 | 30.72±0.78 | **39.03±0.69** |
|  | Tiny-Imagenet | 3.43±0.26 | 4.50±0.04 | 13.75±0.15 | **24.20±0.09** |

Furthermore, we conducted ablation studies for $\tau$, which defines the maximum communication cost allowed per client per round, which determines the minimum fraction of clauses that must remain local ($min\_frac$). It influences only how many parameters can be transmitted per client. We provide experiments with the CIFAR-10 varying $\tau$ in Table C.6 and we observed that at lower heterogeneity levels (Dir(0.1) and Dir(0.05)), increasing $\tau$ (permitting more global clause sharing) yields higher accuracy, since more global knowledge benefits clients that share substantial distributional overlap. However, under extreme heterogeneity (Dir(0.01) and Dir(0.005)), increasing $\tau$ produces only marginal changes as personalization dominates, and additional global clauses offer limited benefit. These results demonstrate that $\tau$ primarily governs the communication budget and does not destabilize or meaningfully alter the personalization behaviour of CS-pFedTM. The accuracy remains stable across a wide range of $\tau$ values, indicating that the clause allocation and masking mechanisms operate consistently regardless of the communication limit.

Table 10: Ablation Studies for $\tau$

| $\tau$ | Dir(0.1) | Dir(0.05) | Dir(0.01) | Dir(0.005) |
|---|---|---|---|---|
| 0.01 | 79.81±0.68 | 86.92±0.83 | 98.79±0.78 | 96.72±0.65 |
| 0.03 | 79.76±0.75 | 86.98±0.69 | 98.85±0.55 | 96.56±0.73 |
| 0.05 | 79.89±0.53 | 87.27±0.63 | 98.54±0.72 | 96.81±0.68 |
| 0.1 | 80.03±0.46 | 87.91±0.85 | 98.89±0.50 | 96.75±0.62 |
| 0.12 | 80.17±0.59 | 88.49±0.68 | 98.75±0.65 | 96.84±0.51 |
| 0.15 | 80.49±0.73 | 88.84±0.51 | 98.77±0.39 | 96.59±0.64 |

### C.7 Justification for Performance-Based Client Selection

In FedTM, **TopK** aggregation selects clients with the largest number of class samples. Under full participation (cross-silo), this repeatedly favors the same clients with the most number of samples, leading to fairness issues and poor representation of clients with more challenging or smaller datasets. To address this, CS-pFedTM selects the Top-K clients based on local validation accuracy rather than sample counts.

This design is motivated by two factors:

- Fairness: accuracy-based selection allows clients with fewer samples—but well-trained local models—to contribute, preventing dominance by a small subset of clients.

- Model quality: local performance is a more reliable indicator of useful updates than dataset size, reducing global bias toward data-rich but under-performing clients.

The choice of K = 2 follows FedTM, which showed that Top-2 aggregation provides the best trade-off between information sharing and communication overhead. Top-1 under-utilizes cross-client information, while larger K gives diminishing returns due to the bit-level TM representation.

We evaluated sample-based vs. performance-based **TopK** on CIFAR-10 under both cross-silo (10 clients) and cross-device (100 clients) configurations, with both full and 0.3 partial participation. As shown in Table 11 below, performance-based Top-K consistently achieves higher mean accuracy, lower or comparable variance, and significantly reduces the over-representation of dominant clients in cross-silo settings.

Table 11: Performance of Sample-based and Performance-based **TopK** on various FL settings

|  | Cross-Silo | Cross-Device | |
| --- | --- | --- | --- |
|  | Full participation | Partial participation | Full participation |
| Sample-based **TopK** | 87.23±0.89 | 85.52±0.98 | 86.35±0.69 |
| Performance-based **TopK** | **87.93±1.07** | **86.92±0.83** | **86.96±0.61** |
| Sample-based **TopK** | 77.84±0.78 | 78.07±0.81 | 75.32±0.53 |
| Performance-based **TopK** | **78.04±1.19** | **79.81±0.68** | **77.65±0.55** |

In cross-silo settings, variance for performance-based **TopK** is slightly higher, which reflects greater inclusivity: more clients are selected over time instead of always the same two. Importantly, accuracy is still consistently higher.We also note that model performance remains stable under different participation rates (full participation vs. partial participation), as shown in Figure 4. This indicates that the effectiveness of the performance-based **TopK** is not sensitive to the participation ratio: even when fewer clients participate, the selected updates remain representative of the overall client population. In other words, the selection mechanism does not overfit to any subset of clients, and the aggregation remains robust across both cross-device and cross-silo settings.

Several prior works have shown that client selection schemes that account for client model quality or utility led to better global performance than naive or sample-count–based selection. For example, Jee Cho et al. (2022) and Lai et al. (2021) demonstrate that incorporating client-side metrics, such as loss, update usefulness, or training reliability. These substantially improves convergence and generalization in federated optimization. Although the specific criteria differ from ours, these works reinforce the broader conclusion that data-quantity–based selection tends to introduce bias, while performance-aware selection results in more informative updates.

A full convergence analysis for non-convex, non-differentiable TM training with selective aggregation is, to our knowledge, still an open problem even for simpler TM setups. Our approach follows the standard FL aggregation pattern (averaging over selected client models), and our experiments show stable convergence across all configurations with low variance across the experiments. We therefore position the performance-based **TopK** as a practically motivated, performance-aware client-selection heuristic, analogous in spirit to utility-based selection schemes studied in prior FL work and support it with empirical evidence rather than a full convergence proof.

### C.8 STABILITY OF PARAMETER SIMILARITY AND USE OF THE REFERENCE ROUND

The reference round is used solely to estimate the parameter similarity that guides clause allocation. Because clients are sampled uniformly at random in every round, including the reference round, the participating clients constitute an unbiased sample of the overall population. Thus, the similarity measured in this round provides a reliable estimator of the system's underlying heterogeneity.

Empirically, we computed the client parameter similarity at every training round and reported its average variance across rounds, for varying client participation rates (0.1, 0.3, 0.5, 1.0) and averaged over three independent random seeds in Table12. Across all datasets and heterogeneity settings, the variance is extremely small, indicating that similarity remains tightly concentrated around the reference-round estimate. Although variance decreases slightly as the participation rate increases,

the reduction is minor, indicating that similarity is already highly stable even under low participation. This further confirms that the reference-round estimate remains reliable regardless of sampling rate.

Table 12: Average Variance of parameter similarity across training rounds

|  | Participation Ratio | SVHN | EMNIST | SC-12 | CIFAR-10 | CIFAR-100 | Tiny-Imagenet |
|---|---|---|---|---|---|---|---|
| Dir(0.05) | 0.1 | 0.0120 | 0.0007 | 0.0029 | 0.0054 | 0.0005 | 0.0006 |
|  | 0.3 | 0.0047 | 0.0006 | 0.0024 | 0.0028 | 0.0004 | 0.0008 |
|  | 0.5 | 0.0029 | 0.0005 | 0.0018 | 0.0023 | 0.0004 | 0.0005 |
|  | 1 | 0.0021 | 0.0005 | 0.0009 | 0.0023 | 0.0003 | 0.0005 |
| Dir(0.1) | 0.1 | 0.0023 | 0.0040 | 0.0047 | 0.0116 | 0.0011 | 0.0013 |
|  | 0.3 | 0.0053 | 0.0028 | 0.0035 | 0.0099 | 0.0009 | 0.0007 |
|  | 0.5 | 0.0015 | 0.0021 | 0.0029 | 0.0058 | 0.0009 | 0.0002 |
|  | 1 | 0.0019 | 0.0011 | 0.0006 | 0.0035 | 0.0009 | 0.0002 |

Moreover, Figure 4 shows that model performance remains stable under different participation rates (full participation vs. partial participation). If the similarity estimate were highly sensitive to which clients participate in any individual round, we would expect substantial divergence in accuracy across participation settings. Instead, accuracy remains nearly unchanged, further indicating that the heterogeneity captured in the reference round is representative of subsequent rounds. We also observe consistently low variance in overall performance across runs, reinforcing that system behavior does not fluctuate meaningfully with changes in the sampled client set.

Regarding dynamic data distributions (concept drift), CS-pFedTM is naturally compatible with such settings: since global parameters are already transmitted every round, the system can simply re-estimate inter-client similarity periodically (eg. every N rounds) and update clause allocation accordingly, without modifying the core algorithm or increasing communication cost.

# D    LIMITATIONS AND FUTURE WORK

While CS-pFedTM delivers strong accuracy under heterogeneity and achieves substantial communication savings, its performance remains fundamentally bounded by the current capabilities of TMs. In centralized settings, TMs can lag behind state-of-the-art DNNs due to information loss from booleanization and limited expressive power of bit-level learning. Although recent advances, such as TM composites (Granmo, 2023), multi-encoding architectures, and emerging variants like Graph TMs (Granmo et al., 2025), are beginning to close this gap, improving centralized TM performance remains a prerequisite for further boosting federated accuracy (How et al., 2025). Future extensions of CS-pFedTM could incorporate these enhanced TM architectures, enabling richer clause representations while maintaining efficiency. Another promising direction is adapting CS-pFedTM to concept drift. As similarity is computed independently of training dynamics, the framework can naturally re-estimate similarity every N rounds and update global/local clause allocation as client distributions evolve. Additionally, integrating elements of Coalesced TMs may help reduce static memory footprint, while adaptive clause sparsification or dynamic clause reduction could further lower runtime memory and latency. These directions offer a path toward more expressive, adaptive, and resource-efficient TM-based personalized FL.

# E ALGORITHMS

## E.1 FEDTM IMPLEMENTATION DETAILS

FedTM is the first FL framework that leverages TM to concurrently optimize communication efficiency and memory utilization. In contrast to FL frameworks employing DNNs, where weight aggregation often involves a straightforward weighted averaging of integer weights, FedTM adopts a distinctive two-step aggregation scheme How et al. (2023), owing to the unique structure of TM as described in Section 3.1.

The first step employs the **TopK** algorithm for bit-based aggregation of the TA states. This method selects $K$ clients based on the confidence of the TA states, giving preference to clients with the top $K$ data size for each specific class. The second step involves the **AverageCW** method, specifically tailored for computing the average of the integer clause weights weighted based on the total sample size of each set of local data. This two-step approach ensures the effective aggregation of information encoded in both the bit-based and integer components of TM.

---

**Algorithm 2 FedTM**

---

1. Initialize global parameters $\mathbf{W}_0, \mathbf{S}_0$ with the same TM architecture and clients inform the server of their local dataset sizes, $|D_j|, j = 1, 2, ...N$

**for** communication round $t = 1, 2, ...T$ **do**

  2. For all participating clients, $J$, train a TM model with the current weights, $W_{t-1}$ on their local dataset, $D_j$, for $e$ epochs

  3. Clients upload their local parameters

  4. Aggregation of clients' parameters

  **for** class $m = 1, 2, ...M$ **do**

    $\mathbf{W}_t[m] \leftarrow$ **AverageCW**$(m, \delta, t)$

    $\mathbf{S}_t[m] \leftarrow$ **TopK**$(m, k, t)$

  5. All clients download the new global parameters: $\mathbf{W}_t, \mathbf{S}_t$

---

**AverageCW**$(m, \delta, t)$:

$\mathbf{W}_t[m] \leftarrow int(\frac{1}{|D|} \sum_{j=1}^{J} |D_j| \mathbf{W}_t^j[m])$

**if** t $> 1$ **then**

  **if** $\forall_{j=1}^{J} \mathbf{W}_t^j[m] = 0$ **then**

    $\mathbf{W}_t[m] \leftarrow \mathbf{W}_{t-1}[m]$   if class $m$ is not seen in round $t$ of training then use previous weights

  **else**

    $\mathbf{W}_t[m] \leftarrow (1 - \delta)\mathbf{W}_{t-1}[m] + \delta\mathbf{W}_t[m]$

**return** $int(\mathbf{W}_t[m])$

**TopK**$(m, k, t)$:

$sorted\_list \leftarrow sort(\forall_{j=1}^{J}|D_j|[m])$

$sorted_k \leftarrow sorted\_list[0 : k]$

$\mathbf{S}_t[m] \leftarrow \bigvee_{j}^{sorted_k} \mathbf{S}_t^j[m]$

**return** $\mathbf{S}_t[m]$

---

## E.2 CS-pFedTM Algorithm

---

**Algorithm 3 CS-pFedTM: Communication-Efficient and Similarity-Driven Personalization with TM**

---

**Input:** Total number of clients $N_c$, total communication rounds $T$, number of clauses per client $n_{\text{clauses}}$, communication budget $\tau$

  **for** round $t = 0, 1, \ldots, T$ **do**

    Server randomly samples $N_t$ clients, $\mathcal{C}_t$

    **if** $t == 0$ **then**

      Clients train a tiny reference TM and upload state parameters

      $\texttt{min\_frac} \leftarrow$ **compute\_min\_frac**

      $JS_{\text{clients}} \leftarrow$ **compute\_client\_similarity**

      $\texttt{local\_frac} \leftarrow \exp\left(-\ln(1/\texttt{min\_frac}) \cdot JS_{\text{clients}}\right)$

      Assign local and global clauses:

$$n_{\text{local}} = \lfloor n_{\text{clauses}} \cdot \texttt{local\_frac} \rfloor, \quad n_{\text{global}} = n_{\text{clauses}} - n_{\text{local}}$$

    **for** each client $n \in \mathcal{C}_t$ **do**

      Client trains local model $L^n$, global model $G^n$

      $L^n, G^n \leftarrow$ **mask\_weights**$(L^n)$, **mask\_weights**$(G^n)$

      Client uploads global parameters $G^n$ to the server

    $G_t \leftarrow$ **aggregate\_global\_models**

    Server updates clients' global TM with $G_t$

  **return** Personalized TMs for each client: $TM^n \in \{G_t, L^n\}$, combined using Equation 3

---

---

**Algorithm 4 compute\_min\_frac**

---

  $\texttt{per\_clause\_size} \leftarrow \frac{ref\_model\_size}{ref\_num\_clauses}$

  $\texttt{max\_global\_clauses} \leftarrow \min\left(\lfloor \frac{\tau}{\texttt{per\_clause\_size}} \rfloor, \frac{n\_clauses}{2}\right)$

  $\texttt{min\_local\_clauses} \leftarrow n\_clauses - \texttt{max\_global\_clauses}$

  Minimum local fraction: $\texttt{min\_frac} \leftarrow \frac{\texttt{min\_local\_clauses}}{n\_clauses}$

  **return** $\texttt{min\_frac}$

---

---

**Algorithm 5 compute\_client\_similarity**

---

  $\texttt{total\_similarity} \leftarrow 0$

  $\texttt{pair\_count} \leftarrow 0$

  **for** $pair$ **in** combinations(len($all\_states$), 2) **do**

    $\texttt{total\_similarity} \leftarrow \texttt{total\_similarity} +$ **JSTest**$(pair[0], pair[1])$

    $\texttt{pair\_count} \leftarrow \texttt{pair\_count} + 1$

  $\texttt{average\_jaccard\_similarity} \leftarrow \frac{\texttt{total\_similarity}}{\texttt{pair\_count}}$ **if** pair\_count $> 0$ **else** $0$

  **return** $\texttt{average\_jaccard\_similarity}$

---

---

**Algorithm 6 JSTest($S^A, S^B$)**

---

**if** $\text{len}(S^A) \neq \text{len}(S^B)$ **then**
    **raise** ValueError("Vectors must have the same length")
intersection $\leftarrow \sum_{i=0}^{\text{len}(S^A)} S^A[i] \bigwedge S^B[i]$
union $\leftarrow \sum_{i=0}^{\text{len}(S^A)} S^A[i] \bigvee S^B[i]$
**if** union $== 0$ **then**
    **return** 0
**else**
    **return** $\frac{\text{intersection}}{\text{union}}$

---

**Algorithm 7 mask_weights($W$)**

---

**for** class $m = 1, 2, ...M$ **do**
    **if** m is not present in local data **then**
        $W[m] = 0$
**return** $W$

---

**Algorithm 8 aggregate_global_models**

---

Input: list of client models $\mathcal{G}_t$, where each $G^n \in \mathcal{G}_t$ contains their weights, $W_t^n$ and states, $S_t^n$
Rank of clients based on performance: $rank\_clients_{t-1}$
Global weights $\mathbf{W}$ and states $\mathbf{S}$
**for** class $m = 1, \ldots, M$ **do**
    $\mathbf{W}_t[m] \leftarrow \mathbf{AverageCW}$
    $\mathbf{S}_t[m] \leftarrow \mathbf{Top2_{Perf}}$
**return** $G_t = \{\mathbf{W}_t, \mathbf{S}_t\}$

---

**Algorithm 9 Top2$_{\mathbf{Perf}}$**

---

**if** $rank\_clients_{t-1} > 1$ **then**
    $\mathbf{S}_t[m] = \mathbf{S}_t^{rank\_clients_{t-1}[0]}[m] \bigvee \mathbf{S}_t^{rank\_clients_{t-1}[1]}[m]$
**else**
    $\mathbf{S}_t[m] = \mathbf{S}_t^{rank\_clients_{t-1}[0]}[m]$
**return** $\mathbf{S}_t[m]$

---

**Use of LLMs:** We used LLMs only at the sentence level (e.g., grammar correction and rewording). No LLMs were used for retrieval, discovery, research ideation, or any other purpose.

