# OpenReview forum: "CS-pFedTM: Communication-Efficient and Similarity-based Personalization with Tsetlin Machines"
_ICLR.cc/2026/Conference — Submitted to ICLR 2026_

### Official Review · Reviewer_jpnA · 2025-10-28

**Soundness:** 2
**Presentation:** 2
**Contribution:** 2
**Rating:** 6
**Confidence:** 3

**Summary:**

This paper addresses the challenges of data heterogeneity and communication efficiency in federated learning (FL). The authors propose CS-pFedTM, a personalized FL method based on Tsetlin Machines (TMs), which uses two independent models per client: a local TM for client-specific patterns and a global TM for shared knowledge. The core contribution is an adaptive clause allocation mechanism that dynamically balances local and global clauses based on client parameter similarity (as a proxy for data heterogeneity) and communication budgets. The method incorporates performance-based client selection and class-specific weight masking. Experiments on image and audio datasets demonstrate improved accuracy, reduced communication costs, and enhanced storage and runtime efficiency compared to state-of-the-art personalized FL baselines.

**Strengths:**

The use of Tsetlin Machines for FL personalization, with adaptive local-global clause allocation based on parameter similarity, is a unique contribution.
The method significantly reduces upload/download costs, storage, and runtime memory while maintaining competitive accuracy
The inverse relationship between parameter similarity (Jaccard index) and data heterogeneity (Wasserstein distance) is formally motivated and empirically validated.

**Weaknesses:**

Only FedTM is used as a TM-based baseline, leaving other potential TM variants or composites unexplored.
Although results are averaged over three runs, no standard deviations or confidence intervals are provided.
The individual contributions of components like weight masking, similarity-driven allocation, and performance-based selection are not isolated.
The method relies on booleanized inputs, which may limit its use in domains where non-binary feature representations are critical.
No discussion of scenarios where parameter similarity may fail to accurately reflect data heterogeneity, or how the method performs under extreme non-IID conditions.
The method uses a reference round for similarity estimation, but dynamic changes in client data distributions are not addressed.
While performance-based selection is used, its impact on fairness or long-term participation bias is not analyzed.

**Questions:**

Please respond to the Weaknesses.

---

> ### Author Response · Authors · 2025-11-19
>
> We appreciate your feedback and would like to address the concerns you raised.
>
> **Comparison with other TM-based FL**
>
> We identified a recent TM-based personalized FL method, TPFL, Tsetlin-Personalized Federated Learning with Confidence-Based Clustering [1], a multi-center personalized FL method for TM that clusters clients based on their confidence and aggregates weights within each cluster, enabling personalized model updates under non-IID data.
>
> We now include TPFL as an additional TM baseline. Using the same number of clauses, CS-pFedTM achieves higher performance across all datasets while requiring lower communication. On average, CS-pFedTM outperforms TPFL by $6.24\%$, and reduces upload and download communication by $3.60\times$ and $9.28\times$. The results are in **Reported Test Accuracy with Standard Deviation**  ([link](https://openreview.net/forum?id=rcY4qPSXP4&noteId=ISL9JEFBWm))
>
> Furthermore, under increasing heterogeneity, where we varied the number of classes per client on the CIFAR-10 dataset, CS-pFedTM maintains superior performance, whereas TPFL degrades more sharply under stronger heterogeneity.
>
> **Table9: Performance of CS-pFedTM vs TPFL on data heterogeneity under different splits, where we vary the number of classes per client**
> |      | CS-pFedTM | TPFL  |
> | ---- | --------- | ----- |
> | C(2) | **85.73±0.88**     | 79.21±0.62|
> | C(3) | **73.52±0.73**     | 65.79±1.18|
> | C(4) | **64.29±1.07**     | 55.21±0.94 |
> | C(5) | **56.33±0.82**     | 48.06±0.77 |
> | C(6) | **51.67±1.12**     | 42.76±0.81 |
>
> [1] Rasoul Jafari Gohari, Laya Aliahmadipour, and Ezat Valipour. TPFL: Tsetlin-Personalized Federated Learning with Confidence-Based Clustering, 2024.
>
> **Clarification on Booleanization**
>
> Training TMs requires Boolean input; however, Booleanization is a standard and well-established preprocessing step in TM-based models [2-3]. To satisfy this requirement while preserving the discriminative structure of each dataset, we apply dataset-specific encoding schemes that have been validated in prior TM work. For EMNIST, pixels are thresholded at 40 (values > 40 =1, otherwise 0). For SVHN, we use adaptive Gaussian thresholding with window size 11 and threshold 2 following [1]. For CIFAR-10, CIFAR-100 and Tiny-ImageNet, we booleanized using 4x4 color thermometer encoding [2]. These preprocessing methods are standard in the TM literature and preserve sufficient structure for competitive performance across image datasets. We have clarified this more explicitly in the revised text.
>
> [2] Ole-Christoffer Granmo. The Tsetlin Machine - A Game Theoretic Bandit Driven Approach to Optimal Pattern Recognition with Propositional Logic, 2021.
>
> [3] Ole-Christoffer Granmo. TMComposites: Plug-and-Play Collaboration Between Specialized Tsetlin Machines, 2023
>
> **Discussion on Limitations of using Parameter Similarity**
>
> We acknowledge that parameter similarity may lose granularity under extreme non-IID conditions. In such cases, the similarity between client clauses can saturate near zero, and the resulting clause allocation becomes highly personalized. This behaviour is expected: when clients' distributions are highly dissimilar, the system should retain only a minimal shared component and prioritize local adaptation.
>
> Importantly, this does not harm performance in practice. Our experiments already include severe non-IID settings (eg. CIFAR-10 with only 2-3 classes per client), and CS-pFedTM still exhibits stable convergence and achieves the highest accuracy among all baselines (Figure 3). This demonstrates that clause similarity remains sufficiently informative to guide personalization even under strong label imbalance.
>
> To further test the method, we evaluated CS-pFedTM under extreme non-IID data partition (Dir(0.01) and Dir(0.005)), where heterogeneity is substantially higher and global sharing naturally becomes minimal. CS-pFedTM continues to perform comparably or better than the baselines, confirming that the similarity-driven mechanism behaves robustly even when the model becomes almost fully personalized.
>
> **Table10: Performance of the algorithms in the extreme non-IID setting**
>
>
> |          | Dir(0.005)       | Dir(0.01)        |
> |-----------------|------------------|------------------|
> | FedAvg          | 28.18±0.74       | 25.36±0.58       |
> | FedAvg++        | 95.91±1.12       | 92.27±0.84       |
> | pfedFDA         | 94.89±0.66       | 91.86±0.91       |
> | FedPAC          | 97.81±0.93       | **96.98±0.77**       |
> | FedRep          | 97.25±0.85       | 95.20±0.64       |
> | FedPer          | 98.04±1.02       | 96.67±0.72       |
> | LG-FedAvg       | 98.57±0.81       | 94.36±1.10       |
> | FedSelect(0.3)  | 98.12±0.69       | 96.20±0.95       |
> | FedSelect(1.0)  | 96.49±0.54       | 95.88±1.16       |
> | TPFL            | 98.38±1.07       | 96.29±0.91       |
> | FedTM           | 16.78±0.48       | 15.93±0.62       |
> | CS-pFedTM       | **98.79±0.78**       | 96.72±0.65      |

---

> > ### Author Response · Authors · 2025-11-19
> >
> > **General Concerns**
> >
> > Regarding the lack of confidence intervals of the reported results: Please see **Reported Test Accuracy with Standard Deviation** ([link](https://openreview.net/forum?id=rcY4qPSXP4&noteId=ISL9JEFBWm))
> >
> > Regarding isolating the contributions of personalization, weight masking: Please see **Ablation Studies** ([link](https://openreview.net/forum?id=rcY4qPSXP4&noteId=O6xzQy90tN))
> >
> > Regarding the of the dynamic changes in client data: Please see **Clarification on Reference Round** ([link](https://openreview.net/forum?id=rcY4qPSXP4&noteId=qC2xMhgZD8))
> >
> > Regarding the fairness and long-term participation bias of performance-based TopK: Please see **Rationale and Benefits of Performance-Based TopK**
> > ([link](https://openreview.net/forum?id=rcY4qPSXP4&noteId=VdvSC7tBKP)) and **Empirical Support and Convergence Behaviour of Performance-Based TopK** ([link](https://openreview.net/forum?id=rcY4qPSXP4&noteId=Q51ZzYcP1w))

---

### Official Review · Reviewer_rMfz · 2025-10-31

**Soundness:** 1
**Presentation:** 2
**Contribution:** 2
**Rating:** 4
**Confidence:** 3

**Summary:**

This paper addresses the data heterogeneity problem of the standard FL framework. The proposed method, CS-pFedTM, introduces Tsetlin Machines, a rule-based model based on finite-state automata and game theory, as an efficient alternative to DNNs, along with communication budget-aware resource allocation and heterogeneity-driven personalization method. Furthermore, the proposed method allocates clauses between local and global models based on client similarity computed from a reference round, and incorporates weight masking and performance-based client selection. Experiments under various models and datasets demonstrate that the proposed method outperforms SOTA personalized FL methods. While the application of TM to personalized FL is novel, the paper suffers from significant weaknesses in theoretical justification and experimental rigor that place it below the acceptance threshold.

**Strengths:**

This paper presents a novel integration of the Tsetlin Machine into a personalized federated learning framework to address data heterogeneity. In addition, complementary schemes are proposed to enhance personalization performance and communication efficiency.

**Weaknesses:**

(1) The proposed method largely depends on empirical extension of existing methods, lacking theoretical justification. To address this concern, please clarify and provide detailed explanations for the following:

(1-a) The paper claims that, to maintain fairness in client participation, only the top-performing clients’ states are uploaded and used in global aggregation (lines 295–298). However, this design may bias the model toward specific data distributions and exclude clients with more challenging datasets, potentially degrading generalization across the population. Moreover, the paper does not provide any theoretical justification or convergence analysis for this selective aggregation approach. Please clarify this concern.
Besides, please provide details of the client selection method: How are the top-performing clients selected? Is it Top-1 or Top-K? If it is Top-K, what is the default value of K, and how is it determined?

(1-b) The paper claims that the class-specific weights of TMs enable further personalization through weight masking: weights corresponding to classes not observed locally can be set to zero, allowing the model to quickly adapt to unseen classes (lines 201-203). However, the paper does not provide sufficient theoretical or empirical evidence to explain how this zero-masking mechanism contributes to fast adaptation. Please clarify the underlying rationale or mechanism — for example, whether zeroing out unobserved class weights reduces interference, facilitates gradient reallocation, or improves generalization.

(1-c) Due to the randomness of participating clients, the communication cost and similarity computed in the reference round may not reflect those in subsequent rounds. How can the authors ensure that the reference round is representative of the remaining rounds under random client participation?

(1-d) The explanation of how min_frac is computed (line 332) is provided in Algorithm 4 of the appendix. However, the paper does not offer any rationale or empirical justification for this computation scheme. Please provide a detailed description of how min_frac is determined and explain the reasoning behind this choice.

(2) The experimental claims do not fully support the contribution of the proposed method. To address this concern, please clarify and provide detailed explanations for the following:

(2-a) All results appear to be from single runs without confidence intervals. For rigorous evaluation, please provide the standard deviation over multiple random seeds.

(2-b) The 2-layer CNN model used in the experiments is not a commonly adopted architecture. At minimum, results on more standard models, such as a 4-layer CNN or MobileNet, are recommended as baselines for fair comparison.

(2-c) To assess the individual contribution of each component of the proposed method, please provide ablation studies where client selection, weight masking, and clause allocation are applied independently.

(2-d) The current presentation of Fig. 3 makes it difficult to visually compare algorithmic performance. It is recommended to present these results in a tabular format for easier comparison and clearer interpretation.

**Questions:**

Please refer to the Weakness section for detailed comments. In particular, I would appreciate clarification on the questions raised for each weakness. I will reconsider my evaluation after reviewing the authors’ rebuttal to these points.

---

> ### Author Response · Authors · 2025-11-19
>
> We appreciate your feedback and would like to address the concerns you raised.
>
> **Clarification on Masking Mechanism**
>
> In a TM, class-specific clause weights are updated through both positive and negative feedback. Even if a client never observes a particular class locally, its weights for that class are still influenced indirectly through negative updates that occur when training on other classes. These weights therefore encode only "not-class-X" evidence and do not contain any meaningful information for that class. Retaining these inherited negative-only weights offers no benefit and may even contradict the client’s actual data distribution, introducing cross-class interference.
>
> Masking addresses this issue by setting all class-specific weights for unseen classes to zero, placing those classes in a completely neutral state. This removes the harmful interference caused by accumulated negative updates for unseen classes and prevents the client from having to "unlearn" incorrect clause–class associations before beginning to learn useful ones. As a result, when the client eventually encounters an unseen class, whether due to heterogeneous data partitions or natural distribution shifts, it can begin learning from a clean baseline rather than correcting inherited noise. This leads to faster and more stable adaptation, as the client’s early updates directly shape the correct clause–class behaviour. In scenarios such as concept drift or delayed class exposure, masking eliminates the unlearning phase entirely, enabling quicker, unobstructed adaptation to new classes.
>
>
> **Clarification on *min_frac***
>
> The purpose of *min_frac* is to enforce a communication-budget lower bound on the number of local clauses, while allowing similarity-based personalization to adapt smoothly to client heterogeneity. Our choice of an exponential mapping is directly motivated by an empirical regularity observed consistently across all datasets.
>
> As shown in Figure 2, the relationship between client data divergence (measured via the Wasserstein distance) and the similarity of their trained TM parameters follows an exponential-like decay. When client distributions are close (low W), parameter similarity decreases only gradually. However, once heterogeneity becomes large (high W), similarity drops very sharply, with the steepest slope appearing in the high-W region.
>
> This behaviour has direct implications for personalization. In highly heterogeneous regions, small differences in parameter similarity correspond to large underlying distribution shifts, so the allocation should adapt more. In regions where clients are already similar, further changes in similarity reflect negligible distributional differences, so the allocation should vary mildly. Therefore, an exponential form captures exactly this behaviour: it is steep when similarity is low and naturally saturates when similarity is high. Setting *c = ln(1/min_frac)* ensures that the personalization fraction never violates the communication budget while still adhering to the observed exponential relationship between heterogeneity and parameter similarity

---

> ### Author Response · Authors · 2025-11-19
>
> **Comparison with Larger baseline models**
>
> Our primary focus is efficiency-oriented personalized FL, where comparisons are typically made under realistic communication and computational constraints. In this setting, lightweight CNNs remain the standard choice across recent personalization literature, as they better reflect practical FL deployments. Nonetheless, to address the concern directly, we additionally evaluated the DNN-based pFL methods using MobileNet-v2 on CIFAR-10; these results have been included in the appendix. The outcomes align with the expected behavior of parameter-decoupled FL methods where methods that personalize a substantial portion of the model (eg. LG-FedAvg) retain reasonable performance even with MobileNet, since a large number of parameters are adapted locally. However, methods that personalize only the classifier head (FedPer, FedRep, FedPAC) perform worse than with a 2-layer CNN. This is expected: MobileNet’s large shared backbone dominates the representation, and personalizing only the final layer is insufficient to overcome strong distribution shifts under heterogeneous data.
> Importantly, even with this much larger backbone model, the communication cost of these MobileNet-based baselines remains several orders of magnitude higher than CS-pFedTM. Despite using a lightweight architecture, CS-pFedTM achieves comparable accuracy while maintaining its primary advantage of reduced communication.
>
> **Table7 Performance of the algorithms with MobileNet as the baseline model**
>
> |                | Dir(0.05)  |           | Dir(0.1)   |           |
> | -------------- | ---------- | --------- | ---------- | --------- |
> |                | Acc        | CC        | Acc        | CC        |
> | FedAvg         | 39.73±1.99 | 268/895   | 33.25±1.34 | 268/895   |
> | FedAvg++       | 72.4±1.31  | "         | 60.53±1.02 | "         |
> | pfedFDA        | 88.01±1.17 | "         | 80.93±0.91 | "         |
> | FedPAC         | 75.29±0.96 | 267/890   | 69.64±0.83 | 267/890   |
> | FedRep         | 80.72±1.25 | "         | 78.44±1.11 | "         |
> | FedPer         | 76.79±0.92 | "         | 66.17±0.73 | "         |
> | LG-FedAvg      | 87.68±1.83 | 6.66/10.2 | **83.21±1.55** | 6.66/10.2 |
> | FedSelect(0.3) | **88.77±0.53** | 7.63/7.63 | 78.76±0.83 | 7.63/7.63 |
> | FedSelect(1.0) | 85.71±0.19 | 25.4/25.4 | 80.18±0.22 | 25.4/25.4 |
> | CS-pFedTM      | 87.34±0.51 | **0.02/0.44** | 80.34±0.96 | **0.26/5.76** |
>
>
> **Presentation of Figure 3**
>
>
> In the revised version, we present Figure 3’s results in table, allowing easier direct comparison across algorithms, datasets, and budgets.
>
> **Table8: Performance of the algorithms for FL with data heterogeneity under different splits, where we vary the number of classes per client**
>
> |                    | C(2)           | C(3)           | C(4)           | C(5)           | C(6)           |
> | ------------------ | -------------- | -------------- | -------------- | -------------- | -------------- |
> | FedAvg            | 27.64±0.73     | 38.18±0.92     | 29.53±0.61     | 31.35±0.88     | 31.20±0.57     |
> | FedAvg++           | 36.83±1.12     | 53.30±0.84     | 23.49±0.77     | 14.93±0.66     | 13.90±0.59     |
> | pfedFDA            | 49.29±0.64     | 47.28±0.93     | 43.96±0.71     | 41.11±1.02     | 40.52±0.88     |
> | FedPAC            | 80.64±0.91     | 66.36±1.14     | 68.59±0.85     | 64.04±0.74     | 60.75±0.97     |
> | FedRep             | 80.16±0.78     | 72.02±1.03     | 63.56±0.66     | 60.04±0.89     | 57.75±0.82     |
> | FedPer           | 75.53±0.95     | 67.75±0.72     | 65.28±0.93     | 60.91±0.68     | 57.16±0.74     |
> | LG-FedAvg          | 70.91±0.87     | 57.83±0.55     | 54.39±1.11     | 49.80±0.63     | 46.52±1.08     |
> | FedSelect 0.3      | 85.35±0.83     | 72.56±0.79     | 63.12±1.04     | 54.23±0.71     | 51.14±0.69     |
> | TPFL             | 79.21±0.62     | 65.79±1.18     | 55.21±0.94     | 48.06±0.77     | 42.76±0.81     |
> | FedTM           | 37.13±0.59     | 36.54±0.68     | 35.80±0.74     | 37.59±0.92     | 36.03±0.57     |
> |CS-pFedTM         | 85.73±0.88     | 73.52±0.73     | 64.29±1.07     | 56.33±0.82     | 51.67±1.12     |
> | CS-pFedTM (high budget) | 85.46±0.74     | 73.90±0.96     | 66.68±0.85     | 58.50±1.04     | 53.88±0.91     |

---

> > ### Author Response · Authors · 2025-11-22
> >
> > **General Concerns**
> >
> > Regarding the TopK Aggregation Scheme: Please see **Rationale and Benefits of Performance-Based TopK**
> > ([link](https://openreview.net/forum?id=rcY4qPSXP4&noteId=VdvSC7tBKP)) and **Empirical Support and Convergence Behaviour of Performance-Based TopK** ([link](https://openreview.net/forum?id=rcY4qPSXP4&noteId=Q51ZzYcP1w))
> >
> > Regarding the Representativeness of the Reference Round: Please see **Clarification on Reference Round** ([link](https://openreview.net/forum?id=rcY4qPSXP4&noteId=qC2xMhgZD8))
> >
> > Regarding the lack of confidence intervals of the reported results: Please see **Reported Test Accuracy with Standard Deviation**  ([link](https://openreview.net/forum?id=rcY4qPSXP4&noteId=ISL9JEFBWm))
> >
> > Regarding isolating the contributions of personalization, weight masking: Please see **Ablation Studies** ([link](https://openreview.net/forum?id=rcY4qPSXP4&noteId=O6xzQy90tN))

---

### Official Review · Reviewer_GfiK · 2025-11-03

**Soundness:** 2
**Presentation:** 3
**Contribution:** 3
**Rating:** 4
**Confidence:** 4

**Summary:**

This paper proposes a novel FL framework called CS-pFedTM. The method personalizes client models by adaptively allocating local and global clauses in a Tsetlin Machine based on inter-client parameter similarity, which serves as a proxy for distributional heterogeneity. The approach integrates class-specific weight masking and performance-based client selection to enhance scalability. Empirical studies on six benchmark datasets demonstrate the communication efficiency and overall performance.

**Strengths:**

This paper studies an interesting and important topic. The algorithm design is clear and novel, and the experimental results show that the proposed method significantly reduces communication overhead while maintaining accuracy.

**Weaknesses:**

* In contribution point 2, “TM parameters reflect overall system heterogeneity”, why is the system heterogeneity being reflected?

* Does the class-specific weight masking raise additional privacy issues or concerns?

* Corollary 1 seems more like an empirical hypothesis than a rigorous theorem in form. I think there are some issues: W $\to$ 0 then J $\to$ 1 only shows that if the distributions are similar, then the parameters are similar. But it does not mean that when the parameters are similar, the distributions are also similar. For example, when facing noisy and nonconvex models, the parameter similarity may still be high. I think only if the author wants to argue that low parameter similarity corresponds to higher distributional divergence would it make more sense. But why not just write that argument?

* Why the design is “but only the top-performing clients’ states (based on local performance) are uploaded and used in global aggregation”? In my view, this design may lose a lot of useful update information from other clients. Moreover, the design of similarity-driven personalization also leads to a trend where the global model is primarily driven by clients that share similar distributions and have high upload volumes, causing it to lose the long-tailed knowledge.

* No experiment isolates the contributions from performance-based client selection or weight masking (mentioned in Sec. 4.1 and Algorithm 1). Also, the ablation related to the masking parameter \tau is missing.

**Questions:**

See detailed questions in the weaknesses part.

---

> ### Author Response · Authors · 2025-11-19
>
> We appreciate your feedback and would like to address the concerns you raised.
>
>
> **Clarification on “System Heterogeneity” in Contribution Point 2**
>
> In Contribution Point 2, our wording “system heterogeneity” was supposed to be data heterogeneity. We do not refer to system heterogeneity in the FL sense (eg. hardware/computation/latency differences across devices). What we intended to state is that TM parameters reflect the data heterogeneity (distributional divergence) across clients. This follows directly from TM learning dynamics: clauses are reinforced only when specific patterns occur locally, so clients with divergent label/feature distributions reinforce different subsets of clauses, leading to lower parameter similarity. Aggregating these parameter similarities across all participating clients yields an empirical estimate of distributional heterogeneity of the overall client population. We have revised the contribution statement and terminology to explicitly refer to data heterogeneity, ensuring that the intended meaning is clear.
>
> **Privacy Concerns**
>
> First, no metadata about label frequencies, or client-specific label distributions is ever transmitted. In contrast, methods such as FedPAC, FedAvg and FedTM communicate class-distribution metadata, which is known to increase privacy exposure. Our masking does not transmit any auxiliary information tied to class identity.  Second, zero-valued clause weights do not signal class absence. In TMs, clause weights can become zero due to normal training dynamics, even for classes that are present. Therefore, preventing the server from inferring a client’s label space. Finally, if stronger guarantees are desired, CS-pFedTM can be combined with lightweight privacy mechanisms after masking is applied locally (eg. adding small-magnitude Differential Privacy noise to the transmitted clause parameters).  Moreover, in heterogeneous (non-IID) settings, model performance is driven primarily by the personalized local clauses; therefore, adding small noise to the shared global clauses weights has negligible impact on accuracy or convergence, while further hiding any residual patterns that could otherwise reveal class presence.
>
> **Clarification on Corollary 1**
>
> We agree that the original formulation of Corollary 1 may have appeared stronger than what our method requires. Our intention is not to claim a theoretical equivalence between distributional similarity and parameter similarity. Instead, the corollary is meant to capture the relationship that is intrinsic to TM training. In a TM, each clause is reinforced only when specific class-or pattern-dependent events occur. Therefore, when two clients’ data distributions diverge, the sets of clauses reinforced on each client diverge as well. This produces increasingly different TM states. Conversely, when distributions are similar, the reinforcement signals received by clauses are also similar, leading to higher overlap in the learned clause sets. Thus, TM training inherently couples distributional divergence with parameter dissimilarity, and this relationship is sufficiently stable to be used as a proxy for estimating heterogeneity in our method. We have revised the corollary wording to remove any implication of a full theoretical equivalence and to better reflect its intended scope as a monotonic, empirically verified relationship.
>
>
> Corollary 1:
>
> Let $q_A$ and $q_B$ be two class distributions and $S_A$, $S_B$ be the
> corresponding trained TM states (sets of clauses). Then:
> $$W(q_A, q_B) \text{ smaller}
> \quad \Longrightarrow \quad
> \mathcal{J}(S_A, S_B) \text{ larger}.$$
>
> Thus, lower distributional divergence corresponds to higher parameter
> similarity.
>
> **General Concerns**
>
> Regarding the rationale for selecting only top-performing clients and concerns about losing updates or long-tailed knowledge: Please see: **Rationale and Benefits of Performance-Based TopK**
> ([link](https://openreview.net/forum?id=rcY4qPSXP4&noteId=VdvSC7tBKP)) and **Empirical Support and Convergence Behaviour of Performance-Based TopK** ([link](https://openreview.net/forum?id=rcY4qPSXP4&noteId=Q51ZzYcP1w))
>
> Regarding isolating the contributions of client selection, weight masking, and the role of τ:
> Please see **Ablation Studies** ([link](https://openreview.net/forum?id=rcY4qPSXP4&noteId=O6xzQy90tN))

---

### Author Response · Authors · 2025-11-19
**Ablation Studies**

We have included the following ablation results in the paper. By isolating the key components of CS-pFedTM, (i) weight masking and (ii) similarity-driven personalization. Each component independently improves accuracy under heterogeneous data, with masking enhancing robustness to locally absent classes and similarity-driven allocation improving adaptability to heterogeneity. However, the joint application yields the best overall performance and stability across. This confirms that both mechanisms contribute complementary benefits rather than redundant effects. We have included these ablation results in the revised version.

**Table1: Ablation Studies for CS-pFedTM**

|           |               | FedTM      | Mask only  | Personalization only | CS-pFedTM  |
| --------- | ------------- | ---------- | ---------- | -------------------- | ----------|
| Dir(0.05) | SVHN          | 55.58±1.13 | 57.63±2.44 |  87.48±0.94      |**89.59±0.78** |
|           | EMNIST        | 62.94±1.87 | 63.57±2.19 | 91.70±1.38             | **94.60±0.37** |
|           | SC-12         | 62.33±0.27 | 63.37±0.87 | 88.82±1.93           | **91.16±2.64** |
|           | CIFAR-10      | 37.86±1.90 | 39.48±0.82 | 85.31±0.91           | **86.92±0.83** |
|           | CIFAR-100     | 4.37±0.06  | 6.61±0.19  | 43.53±0.71           | **48.20±0.85** |
|           | Tiny-Imagenet | 3.67±0.06  | 4.58±0.09  | 21.28±0.58           | **29.25±0.66** |
| Dir(0.1)  | SVHN          | 59.02±3.77 | 59.66±0.84 |  81.83±0.92          | **83.91±1.61** |
|           | EMNIST        | 69.44±1.87 | 71.15±0.42 | 88.49±1.30        | **91.51±0.54** |
|           | SC-12         | 62.37±0.21 | 63.14±0.84 | 91.01±0.07           | **91.76±1.21** |
|           | CIFAR-10      | 39.62±0.31 | 41.98±0.78 | 77.15±0.24           | **79.81±0.68** |
|           | CIFAR-100     | 4.52±0.57  | 9.12±0.49  | 30.72±0.78           | **39.03±0.69** |
|           | Tiny-Imagenet | 3.43±0.26  | 4.50±0.04  | 13.75±0.15           | **24.20±0.09** |


Furthermore, we clarify that $\tau$ does not control weight masking. $\tau$ defines the maximum communication cost allowed per client per round, which determines the minimum fraction of clauses that must remain local (*min_frac*). It influences only how many parameters can be transmitted per client. We provide experiments varying $\tau$ and we observed that at lower heterogeneity levels (Dir(0.1) and Dir(0.05)), increasing $\tau$ (permitting more global clause sharing) yields higher accuracy, since more global knowledge benefits clients that share substantial distributional overlap. However, under extreme heterogeneity (Dir(0.01) and Dir(0.005)), increasing $\tau$ produces only marginal changes as personalization dominates, and additional global clauses offer limited benefit. These results demonstrate that $\tau$ primarily governs the communication budget and does not destabilize or meaningfully alter the personalization behaviour of CS-pFedTM. The accuracy remains stable across a wide range of $\tau$ values, indicating that the clause allocation and masking mechanisms operate consistently regardless of the communication limit. We have included these results in the appendix for clarity.


**Table2: Ablation Studies for $\tau$ on CIFAR-10**

| $\tau$ | Dir(0.1)   | Dir(0.05)  | Dir(0.01)  | Dir(0.005) |
| ------- | ---------- | ---------- | ---------- | ---------- |
| 0.01    | 79.81±0.68 | 86.92±0.83 | 98.79±0.78 | 96.72±0.65 |
| 0.03    | 79.76±0.75 | 86.98±0.69 | 98.85±0.55 | 96.56±0.73 |
| 0.05    | 79.89±0.53 | 87.27±0.63 | 98.54±0.72 | 96.81±0.68 |
| 0.1     | 80.03±0.46 | 87.91±0.85 | 98.89±0.50 | 96.75±0.62 |
| 0.12    | 80.17±0.59 | 88.49±0.68 | 98.75±0.65 | 96.84±0.51 |
| 0.15    | 80.49±0.73 | 88.84±0.51 | 98.77±0.39 | 96.59±0.64 |

---

### Author Response · Authors · 2025-11-19
**Reported Test Accuracy with Standard Deviation**

In the current version of the paper, we report the mean accuracy over three random seeds, but we acknowledge that omitting the standard deviations may cause confusion regarding the statistical stability of the results. We have added the standard deviations for all experiments in the main results and in the appendix.


**Table3: Performance of the algorithms for FL with data heterogeneity and CC - Communication Costs (Upload/Download) for SVHN, EMNIST, and SC-12**
| Method | SVHN (Dir 0.05) |     | SVHN (Dir 0.1) |     | EMNIST (Dir 0.05) |     | EMNIST (Dir 0.1) |     | SC-12 (Dir 0.05) |     | SC-12 (Dir 0.1) |     |
|--------|------------------|-----|------------------|-----|--------------------|-----|--------------------|-----|--------------------|-----|--------------------|-----|
|        | Acc             | CC  | Acc             | CC  | Acc               | CC  | Acc               | CC  | Acc               | CC  | Acc               | CC  |
| FedAvg | 29.16±14.42     | 13/43 | 53.84±2.71     | 13/43 | 71.41±4.30       | 15/50 | 56.31±2.73       | 15/50 | 56.37±2.93       | 42/141 | 65.46±0.67       | 42/141 |
| FedAvg++ | 80.08±2.21   | " | 71.05±3.50       | " | 72.71±0.85         | " | 74.54±0.32         | " | 67.82±2.72         | " | 66.35±4.03         | " |
| pFedFDA | 81.28±2.77     | " | 70.58±3.87       | " | 95.73±0.14         | " | 94.26±0.20         | " | 90.57±0.62         | " | 91.11±1.82         | " |
| FedPAC | 83.03±2.06      | " | 82.96±1.79   | " | **95.77±0.50**     | 14/46 | **94.28±0.17**  | 14/46 | 86.67±4.27         | " | 90.20±1.45         | " |
| FedRep | 80.81±3.13      | " | 81.33±2.53       | " | 78.12±0.17         | " | 78.18±0.32         | " | 88.49±2.46         | " | 82.91±3.10         | " |
| FedPer | 83.27±2.00      | " | 76.12±1.29       | " | 94.37±1.00         | " | 92.68±0.37         | " | 90.82±0.48         | " | 90.97±2.87         | " |
| LG-FedAvg | 84.20±1.96 | 0.67/1.0 | 78.95±1.01 | 0.67/1.0 | 75.80±0.52 | 4.2/6.4 | 75.69±0.08 | 4.2/6.4 | 79.42±2.62 | 0.41/0.62 | 76.72±1.64 | 0.41/0.62 |
| FedSelect (0.3) | 79.51±2.25 | 2.3/2.3 | 67.90±1.54 | 2.3/2.3 | 94.18±0.04 | 2.7/2.7 | 91.51±0.25 | 2.7/2.7 | 91.07±0.19 | 7.5/7.5 | 85.83±0.24 | 7.5/7.5 |
| FedSelect (1.0) | 79.45±1.86 | 7.7/7.7 | 68.88±1.74 | 7.7/7.7 | 94.55±0.48 | 8.9/8.9 | 91.78±0.10 | 8.9/8.9 | 91.16±0.16 | 25/25 | 85.85±0.17 | 25/25 |
| TPFL | 86.64±0.71 | 0.12/2.4 | 80.39±0.88 | 0.12/2.4 | 91.99±0.23 | 0.08/5.8 | 89.05±0.22 | **0.08**/4.4 | 83.94±1.40 | 0.19/7.7 | 79.48±3.17 | 0.22/7.7 |
| FedTM | 55.58±1.13 | 0.33/12 | 59.02±3.77 | 0.33/12 | 62.94±1.87 | 1.4/48 | 69.44±1.87 | 1.4/48 | 62.33±0.27 | 1.2/35 | 62.37±0.21 | 1.2/35 |
| CS-pFedTM | **89.59±0.78** | **0.01/0.28** | **83.91±1.61** | **0.05/0.96** | 94.60±0.37 | **0.02/0.5** | 91.51±0.54 | 0.11/**2.2** | **91.16±2.64** | **0.01/0.35** | **91.76±1.21** | **0.02/0.56** |

---

> ### Author Response · Authors · 2025-11-19
> **Reported Test Accuracy with Standard Deviation**
>
> **Table4: Performance of the algorithms for FL with data heterogeneity and CC - Communication Costs (Upload/Download) for CIFAR-10, CIFAR-100, and Tiny-ImageNet**
>
> | Method | CIFAR-10 (Dir 0.05) |     | CIFAR-10 (Dir 0.1) |     | CIFAR-100 (Dir 0.05) |     | CIFAR-100 (Dir 0.1) |     | Tiny-ImageNet (Dir 0.05) |     | Tiny-ImageNet (Dir 0.1) |     |
> |--------|----------------------|-----|----------------------|-----|------------------------|-----|------------------------|-----|----------------------------|-----|----------------------------|-----|
> |        | Acc                 | CC  | Acc                 | CC  | Acc                   | CC  | Acc                   | CC  | Acc                       | CC  | Acc                       | CC  |
> | FedAvg | 31.23±0.82          | 13/43 | 32.99±0.46        | 13/43 | 6.58±0.04            | 14/48 | 6.82±0.15            | 14/48 | 1.50±0.09                | 16/53 | 1.48±0.05                | 16/53 |
> | FedAvg++ | 79.12±2.75        | " | 67.41±2.33           | " | 43.20±0.26            | " | 34.57±0.91            | " | 19.42±2.32                | " | 14.79±2.40                | " |
> | pFedFDA | 85.60±1.71         | " | 77.05±0.78           | " | 47.03±1.25            | " | 38.47±1.91            | " | 28.03±0.89                | " | 23.22±0.74                | " |
> | FedPAC | 85.28±1.37          | " | 79.17±0.75           | " | 45.46±0.60            | " | 37.11±0.70            | " | 28.60±0.36                | 13/43 | 21.59±0.32                | 13/43 |
> | FedRep | 86.43±1.45          | " | 79.48±1.23           | " | 44.44±1.56            | " | 38.01±1.53            | " | 27.47±0.84                | " | 20.77±0.57                | " |
> | FedPer | 83.76±1.63          | " | 76.13±1.29           | " | 43.02±0.54            | " | 34.66±0.82            | " | 26.27±0.44                | " | 19.89±0.50                | " |
> | LG-FedAvg | 84.22±1.60       | 0.67/1.0 | 75.56±0.52     | 0.67/1.0 | 37.88±1.02        | 6.7/10 | 29.07±0.74          | 6.7/10 | 22.62±1.26                | 13/20 | 15.54±0.33                | 13/20 |
> | FedSelect (0.3) | 85.95±0.46 | 2.3/2.3 | 78.47±0.68      | 2.3/2.3 | 47.75±1.34        | 2.6/2.6 | 37.13±0.61          | 2.6/2.6 | 29.11±0.60                | 2.9/2.9 | 22.42±1.83                | 2.9/2.9 |
> | FedSelect (1.0) | 86.37±0.72 | 7.7/7.7 | 78.81±0.93      | 7.7/7.7 | 47.76±0.45        | 8.6/8.6 | 36.04±0.13          | 8.6/8.6 | 30.02±0.74                | 9.6/9.6 | 22.45±1.26                | 9.6/9.6 |
> | TPFL | 85.10±1.01           | 0.08/16 | 77.97±1.37      | 0.08/16 | 41.72±0.76         | 0.04/5.6 | 31.68±0.48         | 0.04/5.9 | 15.59±0.43                | **0.02**/3.2 | 11.41±0.61                | **0.02**/4.1 |
> | FedTM | 37.86±1.90          | 0.37/15 | 39.62±0.31      | 0.37/15 | 4.37±0.06          | 1.3/46 | 4.52±0.57          | 1.3/46 | 3.67±0.06                 | 1.4/53 | 3.43±0.26                 | 1.4/53 |
> | CS-pFedTM | **86.92±0.83** | **0.03/0.76** | **79.81±0.68** | **0.03/0.99** | **48.20±0.85** | **0.04/0.88** | **39.03±0.69** | **0.04/0.88** | 29.25±0.66 | 0.12/**2.6** | **24.20±0.09** | 0.12/**2.6** |

---

### Author Response · Authors · 2025-11-19
**Rationale and Benefits of Performance-Based TopK**

In the TopK aggregation scheme in FedTM, clients were selected based on the largest number of class samples, which led to repeated selection of the same dominant clients, especially under full client participation (cross-silo), causing fairness and representation issues. Our performance-based selection addresses this by selecting the TopK clients (K = 2) based on local validation accuracy, rather than sample count. This choice is motivated by:
- Fairness: clients with fewer samples but well-trained local models are given equal opportunity to contribute, ensuring representation across diverse data distributions.
- Model Quality: local performance provides a more reliable indicator of update quality than dataset size, preventing the global model from being biased toward data-rich but low-performing clients
This approach improves both fairness and convergence stability, as reflected in reduced variance across clients’ accuracies.

Moreover, the choice of K=2 is inherited from FedTM , which showed that aggregating the TopK client models (in their case based on class sample counts) yields the best trade-off between performance and communication, while Top1 often under-utilizes cross-client information and larger K brings diminishing returns under the bit-level TM representation. In our setting, we adopt K=2 for the same reason: it is the maximum number of clients that still provides good aggregation quality without significantly increasing communication cost.

We evaluated this design on CIFAR-10 under both cross-silo (10 clients) and cross-device (100 clients) settings with both full and partial participation. (0.3 participation ratio) As shown in the table below, performance-based TopK consistently outperforms sample-based TopK, achieving higher mean accuracy and comparably low variance under cross-device setups, and mitigating the overrepresentation effect observed in cross-silo settings.

**Table5: Performance of CS-pFedTM with different TopK schemes**

|                        | Method                 | Cross-silo (Full Participation) | Cross-device (Partial Participation) | Cross-device (Full Participation) |
| ---------------------- | ---------------------- | -------------------------------- | ---------------------------------- | --------------------------------- |
| Dir(0.05)              | sample-based Topk      | 87.23±0.89                       | 85.52±0.78                         | 86.35±0.69                        |
|                        | performance-based Topk | **87.93±1.07**                   | **86.92±0.83**                     | **86.96±0.61**                    |
| Dir(0.1)               | sample-based Topk      | 77.84±0.78                       | 78.07±0.81                         | 75.32±0.53                        |
|                        | performance-based Topk | **78.04±1.14**                   | **79.81±0.68**                     | **77.65±0.55**                    |

In cross-silo settings, variance of performance-based TopK is slightly higher, which reflects greater inclusivity: more clients are selected over time instead of always the same two. Importantly, accuracy is still consistently higher.We also note that model performance remains stable under different participation rates (full participation vs. partial participation), as shown in Fig. 4(b). This indicates that the effectiveness of the performance-based TopK is not sensitive to the participation ratio: even when fewer clients participate, the selected updates remain representative of the overall client population. In other words, the selection mechanism does not overfit to any subset of clients, and the aggregation remains robust across both cross-device and cross-silo settings.

---

> ### Author Response · Authors · 2025-11-19
> **Empirical Support and Convergence Behaviour of Performance-Based TopK**
>
> Several prior works have shown that client selection schemes that account for client model quality or utility led to better global performance than naive or sample-count–based selection. For example, [1] and [2] demonstrate that incorporating client-side metrics, such as loss, update usefulness, or training reliability.  These substantially improves convergence and generalization in federated optimization. Although the specific criteria differ from ours, these works reinforce the broader conclusion that data-quantity–based selection tends to introduce bias, while performance-aware selection results in more informative updates.
>
> A full convergence analysis for non-convex, non-differentiable TM training with selective aggregation is, to our knowledge, still an open problem even for simpler TM setups. Our approach follows the standard FL aggregation pattern (averaging over selected client models), and our experiments show stable convergence across all configurations with low variance across the experiments. We therefore position the performance-based TopK as a practically motivated, performance-aware client-selection heuristic, analogous in spirit to utility-based selection schemes studied in prior FL work and support it with empirical evidence rather than a full convergence proof.
>
> [1] Yae Jee Cho, Jianyu Wang, and Gauri Joshi. Towards Understanding Biased Client Selection in Federated Learning . Proceedings of The 25th International Conference on Artificial Intelligence and Statistics
>
> [2] Fan Lai, Xiangfeng Zhu, Harsha V. Madhyastha, and Mosharaf Chowdhury. Efficient Federated Learning via Guided Participant Selection. USENIX Symposium on Operating Systems Design and Implementation,(OSDI), 2021.

---

### Author Response · Authors · 2025-11-19
**Clarification on Reference Round**

The reference round is used solely to estimate the parameter similarity that guides clause allocation. Because clients are sampled uniformly at random in every round, including the reference round, the participating clients constitute an unbiased sample of the overall population. Thus, the similarity measured in this round provides a reliable estimator of the system’s underlying heterogeneity.

Empirically, we computed the client parameter similarity at every training round and reported its average variance across rounds, for varying client participation rates (0.1, 0.3, 0.5, 1.0) and averaged over three independent random seeds. Across all datasets and heterogeneity settings, the variance is extremely small, indicating that similarity remains tightly concentrated around the reference-round estimate. Although variance decreases slightly as the participation rate increases, the reduction is minor, indicating that similarity is already highly stable even under low participation. This further confirms that the reference-round estimate remains reliable regardless of sampling rate. We have included this discussion in the appendix.

**Table6. Average variance of parameter similarity across training rounds (averaged over 3 seeds)**

|           | Participation Ratio | SVHN   | EMNIST | SC-12  | CIFAR-10 | CIFAR-100 | Tiny-Imagenet |
| --------- | ------------------- | ------ | ------ | ------ | -------- | --------- | ------------- |
| Dir(0.05) | 0.1                 | 0.0120 | 0.0007 | 0.0029 | 0.0054   | 0.0005    | 0.0006        |
|           | 0.3                 | 0.0047 | 0.0006 | 0.0024 | 0.0028   | 0.0004    | 0.0008        |
|           | 0.5                 | 0.0029 | 0.0005 | 0.0018 | 0.0023   | 0.0004    | 0.0005        |
|           | 1                   | 0.0021 | 0.0005 | 0.0009 | 0.0023   | 0.0003    | 0.0005        |
|           | 0.1                 | 0.0023 | 0.0040 | 0.0047 | 0.0116   | 0.0011    | 0.0013        |
|Dir(0.1)  | 0.3                 | 0.0053 | 0.0028 | 0.0035 | 0.0099   | 0.0009    | 0.0007        |
|           | 0.5                 | 0.0015 | 0.0021 | 0.0029 | 0.0058   | 0.0009    | 0.0002        |
|           | 1                   | 0.0019 | 0.0011 | 0.0006 | 0.0035   | 0.0009    | 0.0002        |


Moreover, Figure 4(b) shows that model performance remains stable under different participation rates (full participation vs. partial participation). If the similarity estimate were highly sensitive to which clients participate in any individual round, we would expect substantial divergence in accuracy across participation settings. Instead, accuracy remains nearly unchanged, further indicating that the heterogeneity captured in the reference round is representative of subsequent rounds. We also observe consistently low variance in overall performance across runs, reinforcing that system behaviour does not fluctuate meaningfully with changes in the sampled client set.

Regarding dynamic data distributions (concept drift), we agree this is an important extension. CS-pFedTM is naturally compatible with such settings: since global parameters are already transmitted every round, the system can simply re-estimate inter-client similarity periodically (eg. every N rounds) and update clause allocation accordingly, without modifying the core algorithm or increasing communication cost. We have added this discussion to the paper and highlight that CS-pFedTM’s communication-efficient design makes it particularly well-suited for incremental adaptation under concept drift, which we plan to explore in future work.

---

### Author Response · Authors · 2025-11-28
**Follow-up on Rebuttal**

Dear Reviewers,

Thank you again for the thoughtful feedback on our submission. We have addressed all raised concerns in our rebuttal, including additional clarifications and new experiments.

We kindly invite you to revisit the rebuttal and updated results, and we would be grateful for any further thoughts you may have.

Thank you for your time and consideration.

Best regards,

Authors

---

### Meta-Review · Area_Chair_S4bK · 2026-01-08

**Summary:**

The paper proposes CS-pFedTM, a personalized federated learning framework based on Tsetlin Machines. While reviewers found the application of Tsetlin Machines to federated learning novel and the empirical results promising, they raised several significant concerns that collectively justify a reject decision:

Multiple reviewers noted the paper's heavy reliance on empirical extensions without sufficient theoretical justification.

Reviewers identified several problems with the experimental evaluation: Use of non-standard model architectures without comparison to more common baselines; Incomplete ablation studies to isolate the contributions of key components.

Reviewers questioned several design choices, such as the representativeness of the reference round under random client participation, Limited exploration of TM-based alternatives beyond FedTM, etc.

**Reviewer Concerns:**

Addressed Concerns:
The authors provided comprehensive standard deviations for all experiments in the rebuttal.

Added TPFL as another TM-based baseline and included MobileNet comparisons.

Provided ablation results isolating weight masking and similarity-driven personalization.

Outstanding Concerns:

Despite clarifications, the theoretical justification remains weak. No convergence analysis was provided for the performance-based selection scheme.

The authors provided an empirical comparison of performance-based vs. sample-based TopK, but no theoretical or long-term analysis of how performance-based selection affects fairness, particularly for clients with consistently lower performance.

**Reviewer Scores:**

Reviewer GfiK (initial: 4): Likely maintain 4.

Reviewer rMfz (initial: 4): Likely maintain 4.

Reviewer jpnA (initial: 6): Likely maintain 6.

---

### Decision · Program_Chairs · 2026-01-26

Reject